# Scalable Batch-Mode Deep Bayesian Active Learning via Equivalence Class Annealing

**Renyu Zhang[1], Aly A. Khan[2,3], Robert L. Grossman[1,4], Yuxin Chen[1]**

[1]Department of Computer Science, University of Chicago
[2]Department of Pathology, University of Chicago
[3]Department of Family Medicine, University of Chicago
[4]Department of Medicine, University of Chicago
`{zhangr,aakhan,rgrossman1,chenyuxin}@uchicago.edu`

## Abstract

Active learning has demonstrated data efficiency in many fields. Existing active learning algorithms, especially in the context of batch-mode deep Bayesian active models, rely heavily on the quality of uncertainty estimations of the model, and are often challenging to scale to large batches. In this paper, we propose Batch-BALanCe, a scalable batch-mode active learning algorithm, which combines insights from decision-theoretic active learning, combinatorial information measure, and diversity sampling. At its core, Batch-BALanCe relies on a novel decision-theoretic acquisition function that facilitates differentiation among different *equivalence classes*. Intuitively, each equivalence class consists of hypotheses (e.g., posterior samples of deep neural networks) with similar predictions, and Batch-BALanCe adaptively adjusts the size of the equivalence classes as learning progresses. To scale up the computation of queries to large batches, we further propose an efficient batch-mode acquisition procedure, which aims to maximize a novel *information measure* defined through the acquisition function. We show that our algorithm can effectively handle realistic multi-class classification tasks, and achieves compelling performance on several benchmark datasets for active learning under both low- and large-batch regimes. Reference code is released at https://github.com/zhangrenyuuchicago/BALanCe.

## 1 Introduction

Active learning (AL) (Settles, 2012) characterizes a collection of techniques that efficiently select data for training machine learning models. In the *pool-based* setting, an active learner selectively queries the labels of data points from a pool of unlabeled examples and incurs a certain cost for each label obtained. The goal is to minimize the total cost while achieving a target level of performance. A common practice for AL is to devise efficient surrogates, aka *acquisition functions*, to assess the effectiveness of unlabeled data points in the pool.

There has been a vast body of literature and empirical studies (Huang et al., 2010; Houlsby et al., 2011; Wang & Ye, 2015; Hsu & Lin, 2015; Huang et al., 2016; Sener & Savarese, 2017; Ducoffe & Precioso, 2018; Ash et al., 2019; Liu et al., 2020; Yan et al., 2020) suggesting a variety of heuristics as potential acquisition functions for AL. Among these methods, *Bayesian Active Learning by Disagreement* (BALD) (Houlsby et al., 2011) has attained notable success in the context of deep Bayesian AL, while maintaining the expressiveness of Bayesian models (Gal et al., 2017; Janz et al., 2017; Shen et al., 2017). Concretely, BALD relies on a *most informative selection* (MIS) strategy—a classical heuristic that dates back to Lindley (1956)—which greedily queries the data point exhibiting the maximal *mutual information* with the model parameters at each iteration. Despite the overwhelming popularity of such heuristics due to the algorithmic simplicity (MacKay, 1992; Chen et al., 2015; Gal & Ghahramani, 2016), the performance of these AL algorithms unfortunately is *sensitive* to the quality of uncertainty estimations of the underlying model, and it remains an open problem in deep AL to accurately quantify the model uncertainty, due to limited access to training data and the challenge of posterior estimation.

In figure 1, we demonstrate the potential issues of MIS-based strategies introduced by inaccurate posterior samples from a Bayesian Neural Network (BNN) on a multi-class classification dataset. Here, the samples (i.e. hypotheses) from the model posterior are grouped into *equivalence classes* (ECs) (Golovin et al., 2010) according to the Hamming distance between their predictions as shown in figure 1a. Informally, an equivalence class contains hypotheses that are close in their predictions for a randomly selected set of examples (See section 2.2 for its formal definition). We note from figure 1b that the probability mass of the models sampled from the BNN is centered around the mode of the approximate posterior distri-

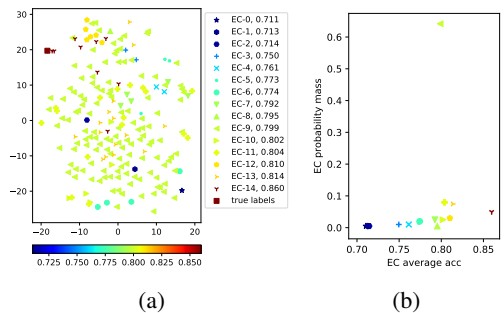

Figure 1: (a) Samples from posterior BNN via MC dropout. The embeddings are generated by applying t-SNE on the hypotheses' predictions on a random hold-out dataset. Colorbar indicates the (approximate) test accuracy of the sampled neural networks on the MNIST dataset. See section C.2 for details of the experimental setup. (b) Probability mass (y-axis) of equivalence classes (sorted by the average accuracy of the enclosed hypotheses as the x-axis).

bution, while little coverage is seen on models of higher accuracy. Consequently, MIS tends to select data points that reveal the maximal information w.r.t. the *sampled distribution*, rather than guiding the active learner towards learning high accuracy models.

In addition to the *robustness* concern, another challenge for deep AL is the *scalability* to large batches of queries. In many real-world applications, fully sequential data acquisition algorithms are often undesirable especially for large models, as model retraining becomes the bottleneck of the learning system (Mittal et al., 2019; Ostapuk et al., 2019). Due to such concerns, batch-mode algorithms are designed to reduce the computational time spent on model retraining and increase labeling efficiency. Unfortunately, for most acquisition functions, computing the optimal batch of queries function is NP-hard (Chen & Krause, 2013); when the evaluation of the acquisition function is expensive or the pool of candidate queries is large, it is even computationally challenging to construct a batch greedily (Gal et al., 2017; Kirsch et al., 2019; Ash et al., 2019). Recently, efforts in scaling up batch-mode AL algorithms often involve diversity sampling strategies (Sener & Savarese, 2017; Ash et al., 2019; Citovsky et al., 2021; Kirsch et al., 2021a). Unfortunately, these diversity selection strategies either ignores the downstream learning objective (e.g., using clustering as by (Citovsky et al., 2021)), or inherit the limitations of the sequential acquisition functions (e.g., sensitivity to uncertainty estimate as elaborated in figure 1 (Kirsch et al., 2021a)).

Motivated by these two challenges, this paper aims to simultaneously (1) mitigate the limitations of uncertainty-based deep AL heuristics due to inaccurate uncertainty estimation, and (2) enable efficient computation of batches of queries at scale. We propose Batch-BALANCE—an efficient batch-mode deep Bayesian AL framework—which employs a decision-theoretic acquisition function inspired by Golovin et al. (2010); Chen et al. (2016). Concretely, Batch-BALANCE utilizes BNNs as the underlying hypotheses, and uses Monte Carlo (MC) dropout (Gal & Ghahramani, 2016; Kingma et al., 2015) or Stochastic gradient Markov Chain Monte Carlo (SG-MCMC) (Welling & Teh, 2011; Chen et al., 2014; Ding et al., 2014; Li et al., 2016a) to estimate the model posterior. It then selects points that can most effectively tell apart hypotheses from different equivalence classes (as illustrated in figure 1). Intuitively, such disagreement structure is induced by the pool of unlabeled data points; therefore our selection criterion takes into account the informativeness of a query with respect to the target models (as done in BALD), while putting less focus on differentiating models with little disagreement on target data distribution. As learning progresses, Batch-BALANCE adaptively anneals the radii of the equivalence classes, resulting in selecting more "difficult examples" that distinguish more similar hypotheses as the model accuracy improves (section 3.1).

When computing queries in small batches, Batch-BALANCE employs an importance sampling strategy to efficiently compute the expected gain in differentiating equivalence classes for a batch of examples and chooses samples within a batch in a greedy manner. To scale up the computation of queries to large batches, we further propose an efficient batch-mode acquisition procedure, which aims to maximize a novel *combinatorial information measure* (Kothawade et al., 2021) defined through our novel acquisition function. The resulting algorithm can efficiently scale to realistic batched learning tasks with reasonably large batch sizes (section 3.2, section 3.3, appendix B).

Finally, we demonstrate the effectiveness of variants of Batch-BALANCE via an extensive empirical study, and show that they achieve compelling performance—sometimes by a large margin—on several benchmark datasets (section 4, appendix D) for both small and large batch settings.

## 2 BACKGROUND AND PROBLEM SETUP

In this section, we introduce useful notations and formally state the (deep) Bayesian AL problem. We then describe two important classes of existing AL algorithms along with their limitations, as a warm-up discussion before introducing our algorithm in section 3.

### 2.1 PROBLEM SETUP

**Notations** We consider pool-based Bayesian AL, where we are given an unlabelled dataset $\mathcal{D}_{\text{pool}}$ drawn *i.i.d.* from some underlying data distribution. Further, assume a labeled dataset $\mathcal{D}_{\text{train}}$ and a set of hypotheses $\mathcal{H} = \{h_1, \ldots, h_n\}$. We would like to distinguish a set of (unknown) target hypotheses among the ground set of hypotheses $\mathcal{H}$. Let $H$ denote the random variable that represents the target hypotheses. Let $p(H)$ be a prior distribution over the hypotheses. In this paper, we resort to BNN with parameters $\omega \sim p(\omega \mid \mathcal{D}_{\text{train}})$[1].

**Problem statement** An AL algorithm will select samples from $\mathcal{D}_{\text{pool}}$ and query labels from experts. The experts will provide label $y$ for given query $x \in \mathcal{D}_{\text{pool}}$. We assume labeling each query $x$ incurs a unit cost. Our goal is to find an adaptive policy for selecting samples that allows us to find a hypotheses with target error rate $\sigma \in [0, 1]$ while minimizing the total cost of the queries. Formally, a *policy* $\pi$ is a mapping $\pi$ from the labeled dataset $\mathcal{D}_{\text{train}}$ to samples in $\mathcal{D}_{\text{pool}}$. We use $\mathcal{D}_{\text{train}}^{\pi}$ to denote the set of examples chosen by $\pi$. Given the labeled dataset $\mathcal{D}_{\text{train}}^{\pi}$, we define $p_{\text{ERR}}(\pi)$ as the expected error probability w.r.t. the posterior $p(\omega \mid \mathcal{D}_{\text{train}}^{\pi})$. Let the cost of a policy $\pi$ be $\text{cost}(\pi) \triangleq \max |\mathcal{D}_{\text{train}}^{\pi}|$, i.e., the maximum number of queries made by policy $\pi$ over all possible realizations of the target hypothesis $H \in \mathcal{H}$. Given a tolerance parameter $\sigma \in [0, 1]$, we seek a policy with the minimal cost, such that upon termination, it will get expected error probability less than $\sigma$. Formally, we seek $\arg\min_{\pi} \text{cost}(\pi)$, s.t. $p_{\text{ERR}}(\pi) \leq \sigma$.

### 2.2 THE EQUIVALENCE-CLASS-BASED SELECTION CRITERION

As alluded in section 1 and figure 1, the MIS strategy can be ineffective when the samples from the model posterior are heavily biased and cluttered toward sub-optimal hypotheses. We refer the readers to appendix A.1 for details of a stylized example where a MIS-based strategy (such as BALD) can perform arbitrarily worse than the optimal policy. A "smarter" strategy would instead leverage the structure of the hypothesis space induced by the underlying (unlabeled) pool of data points. In fact, this idea connects to an important problem for approximate AL, which is often cast as learning *equivalence classes* (Golovin et al., 2010):

**Definition 2.1 (Equivalence Class)** *Let $(\mathcal{H}, d)$ be a metric space where $\mathcal{H}$ is a hypothesis class and $d$ is a metric. For a given set $\mathcal{V} \subseteq \mathcal{H}$ and centers $\mathcal{S} = \{s_1, ..., s_k\} \subseteq \mathcal{V}$ of size $k$, let $r^{\mathcal{S}} : \mathcal{V} \to [k]$ be a partition function over $\mathcal{V}$ and $\mathcal{D}_i := \{h \in \mathcal{V} \mid r^{\mathcal{S}}(h) = i\}$, such that $\forall i, j \in [k], r^{\mathcal{S}}(s_i) = i$ and $\forall h \in \mathcal{D}_i, d(h, s_i) \leq d(h, s_j)$. Each $\mathcal{D}_i \subseteq \mathcal{V}$ is called an equivalence class induced by $s_i \in \mathcal{S}$.*

Consider a pool-based AL problem with hypothesis space $\mathcal{H}$, a sampled set $\mathcal{V} \subseteq \mathcal{H}$, and an unlabeled dataset $\bar{\mathcal{D}}_{\text{pool}}$ which is drawn i.i.d. from the underlying data distribution. Each hypothesis $h \in \mathcal{H}$ can be represented by a vector $v_h$ indicating the predictions of all samples in $\bar{\mathcal{D}}_{\text{pool}}$. We can construct equivalence classes with the Hamming distance, which is denoted as $d_{\text{H}}(h, h')$, and equivalence class number $k$ on sampled hypotheses $\mathcal{V}$. Let $d_{\text{H}}^{\mathcal{S}}(\mathcal{V}) := \max_{h,h' \in \mathcal{V}: r^{\mathcal{S}}(h) = r^{\mathcal{S}}(h')} d_{\text{H}}(h, h')$ be the maximal diameter of equivalence classes induced by $\mathcal{S}$. Therefore, the error rates of any unordered pair of hypotheses $\{h, h'\}$ that lie in the same equivalence class are at most $d_{\text{H}}^{\mathcal{S}}(\mathcal{V})$ away from each other. If we construct the $k$ equivalence-class-inducing centers (as in definition 2.1) as the solution of the max-diameter clustering problem: $\mathcal{C} = \arg\min_{|\mathcal{S}|=k} d_{\text{H}}^{\mathcal{S}}(\mathcal{V})$, we can obtain the minimal worst-case relative error (i.e. difference in error rate) between hypotheses pair $\{h, h'\}$ that lie in the same

---

[1]We use the conventional notation $\omega$ to represent the parameters of a BNN, and use $\omega$ and $h$ interchangeably to denote a hypothesis.

equivalence class. We denote $\mathcal{E} = \{\{h, h'\} : r^{\mathcal{C}}(h) \neq r^{\mathcal{C}}(h')\}$ as the set of all (unordered) pairs of hypotheses (i.e. undirected edges) corresponding to different equivalence classes with centers in $\mathcal{C}$.

**Limitation of existing EC-based algorithms**  Existing EC-based AL algorithms (e.g., EC$^2$ (Golovin et al., 2010) as described in appendix A.2 and ECED (Chen et al., 2016) as in appendix A.3) are not directly applicable to deep Bayesian AL tasks. This is because computing the acquisition function (equation 4 and equation 5) needs to integrate over the hypotheses space, which is intractable for large models (such as deep BNN). Moreover, it is nontrivial to extend to batch-mode setting since the number of possible candidate batches and the number of label configurations for the candidate batch grows exponentially with the batch size. Therefore, we need efficient approaches to approximate the ECED acquisition function when dealing with BNNs in both fully sequential setting and batch-mode setting.

## 3 OUR APPROACH

We first introduce our acquisition function for the sequential setting, namely BALanCe (as in Bayesian Active Learning via Equivalence Class Annealing), and then present the batch-mode extension under both small and large batch-mode AL settings.

### 3.1 THE BALanCe ACQUISITION FUNCTION

We resort to Monte Carlo method to estimate the acquisition function. Given all available labeled samples $\mathcal{D}_{\mathrm{train}}$ at each iteration, hypotheses $\omega$ are sampled from the BNN posterior. We instantiate our methods with two different BNN posterior sampling approaches: MC dropout (Gal & Ghahramani, 2016) and cSG-MCMC (Zhang et al., 2019). MC dropout is easy to implement and scales well to large models and datasets very efficiently (Kirsch et al., 2019; Gal & Ghahramani, 2016; Gal et al., 2017). However, it is often poorly calibrated (Foong et al., 2020; Fortuin et al., 2021). cSG-MCMC is more practical and indeed has high-fidelity to the true posterior (Zhang et al., 2019; Fortuin et al., 2021; Wenzel et al., 2020).

In order to determine if there is an edge $\{\hat{\omega}, \hat{\omega}'\}$ that connects a pair of sampled hypotheses $\hat{\omega}, \hat{\omega}'$ (i.e., if they are in different equivalence classes), we calculate the Hamming distance $d_{\mathrm{H}}(\hat{\omega}, \hat{\omega}')$ between the predictions of $\hat{\omega}, \hat{\omega}'$ on the unlabeled dataset $\tilde{\mathcal{D}}_{\mathrm{pool}}$. If the distance is greater than some threshold $\tau$, we consider the edge $\{\hat{\omega}, \hat{\omega}'\} \in \hat{\mathcal{E}}$; otherwise not. We define the acquisition function of BALanCe for a set $x_{1:b} \triangleq \{x_1, ..., x_b\}$ as:

$$\Delta_{\mathrm{BALanCe}}(x_{1:b} \mid \mathcal{D}_{\mathrm{train}}) \triangleq \mathbb{E}_{y_{1:b}} \mathbb{E}_{\omega, \omega' \sim \mathrm{p}(\omega \mid \mathcal{D}_{\mathrm{train}})} \mathbb{1}_{d_{\mathrm{H}}(\omega, \omega') > \tau} \cdot (1 - \lambda_{\omega, y_{1:b}} \lambda_{\omega', y_{1:b}}) \quad (1)$$

where $\lambda_{\omega, y_{1:b}} \triangleq \frac{\mathrm{p}(y_{1:b} \mid \omega, x_{1:b})}{\max_{y'_{1:b}} \mathrm{p}(y'_{1:b} \mid \omega, x_{1:b})}$ is the likelihood ratio[2] (Chen et al., 2016), and $\mathbb{1}_{d_{\mathrm{H}}(\hat{\omega}_k, \hat{\omega}'_k) > \tau}$ is the indicator function. We can adaptively anneal $\tau$ by setting $\tau$ proportional to BNN's validation error rate $\varepsilon$ in each AL iteration.

In practice, we cannot directly compute equation 1; instead we estimate it with sampled BNN posteriors: We first acquire K pairs of BNN posterior samples $\{\hat{\omega}, \hat{\omega}'\}$. The Hamming distances $d_{\mathrm{H}}(\hat{\omega}, \hat{\omega}')$ between these pairs of BNN posterior samples are computed. Next, we calculate the weight discount factor $1 - \lambda_{\hat{\omega}_k, y_{1:b}} \lambda_{\hat{\omega}'_k, y_{1:b}}$ for each possible label $y$ and each pair $\{\hat{\omega}, \hat{\omega}'\}$ where $d_{\mathrm{H}}(\hat{\omega}, \hat{\omega}') > \tau$. At last, we take the expectation of the discounted weight over all $y_{1:b}$ configurations. In summary, $\Delta_{\mathrm{BALanCe}}(x_{1:b})$ is approximated as

$$\frac{1}{2K^2} \sum_{y_{1:b}} \sum_{k=1}^{K} \left( \mathrm{p}(y_{1:b} \mid \hat{\omega}_k) + \mathrm{p}(y_{1:b} \mid \hat{\omega}'_k) \right) \sum_{k=1}^{K} \mathbb{1}_{d_{\mathrm{H}}(\hat{\omega}_k, \hat{\omega}'_k) > \tau} \left( 1 - \lambda_{\hat{\omega}_k, y_{1:b}} \lambda_{\hat{\omega}'_k, y_{1:b}} \right). \quad (2)$$

$\mathcal{D}_{\mathrm{train}}$ is omitted for simplicity of notations. Note that in our algorithms we never explicitly construct equivalence classes on BNN posterior samples, due to the fact that (1) it is intractable to find the exact solution for the max-diameter clustering problem and (2) an explicit partitioning of the hypotheses samples tends to introduce "unnecessary" edges where the incident hypotheses are

---

[2]The likelihood ratio is used here (instead of the likelihood) so that the contribution of "non-informative examples" (e.g., $\mathrm{p}(y'_{1:b} \mid \omega, x_{1:b}) = \mathrm{const} \ \forall y'_{1:b}, \omega$) is zeroed out.

closeby (e.g., if a pair of hypotheses lie on the adjacent edge between two hypothesis partitions), and therefore may overly estimate the utility of a query. Nevertheless, we conducted an empirical study of a variant of BALANCE with explicit partitioning (which underperforms BALANCE). We defer detailed discussion on this approach, as well as empirical study, to the appendix D.4.

---

**Algorithm 1** Active selection w/ Batch-BALANCE

1: **input**: $\mathcal{D}_{\text{pool}}, \bar{\mathcal{D}}_{\text{pool}}$, aquisition batch size $B$, coldness parameter $\beta$, threshold $\tau$, and downsampling subset size $|\mathcal{C}|$.
2: draw $K$ random pairs of BNN posterior samples $\{\hat{\omega}_k, \hat{\omega}'_k\}_{k=1}^K$
3: **if** $B$ is sufficiently small (see section 4.1) **then**
4:     $\mathcal{A}_B \leftarrow \text{GreedySelection}(\mathcal{D}_{\text{pool}}, \bar{\mathcal{D}}_{\text{pool}}, \{\hat{\omega}_k, \hat{\omega}'_k\}_{k=1}^K, \tau, B)$          # see section 3.2
5: **else**
6:     downsample subset $\mathcal{C} \subset \mathcal{D}_{\text{pool}}$ with $\text{p}(x) \sim \Delta_{\text{BALANCE}}(x)^\beta$
7:     $\mathcal{S}_{1:B}, \mu_{1:B} \leftarrow \text{BALANCE-Clustering}(\mathcal{C}, \bar{\mathcal{D}}_{\text{pool}}, \{\hat{\omega}_k, \hat{\omega}'_k\}_{k=1}^K, \tau, \beta, B)$   # see section 3.3
8:     $\mathcal{A}_B \leftarrow \mu_{1:B}$
9: **output**: $\mathcal{A}_B$

---

In the fully sequential setting, we choose one sample $x$ with top $\Delta_{\text{BALANCE}}(x)$ in each AL iteration. In the batch-mode setting, we consider two strategies for selecting samples within a batch: greedy selection strategy for small batches and acquisition-function-driven clustering strategy for large batches. We refer to our full algorithm as Batch-BALANCE (algorithm 1) and expand on the batch-mode extensions in the following two subsections.

## 3.2 GREEDY SELECTION STRATEGY

To avoid the combinatorial explosion of possible batch number, the greedy selection strategy selects sample $x$ with maximum $\Delta_{\text{BALANCE}}(x_{1:b-1} \cup \{x\})$ in the $b$-th step of a batch. However, the configuration $y_{1:b}$ of a subset $x_{1:b}$ expands exponentially with subset size $b$. In order to efficiently estimate $\Delta_{\text{BALANCE}}(x_{1:b})$, we employ an importance sampling method. The current $M$ configuration samples of $y_{1:b}$ are drawn by concatenating previous drawn $M$ samples of $y_{1:b-1}$ and $M$ samples of $y_b$ (samples drawn from proposal distribution). The pseudocode for the greedy selection strategy is provided in algorithm 2. We refer the readers to appendix B.2 for details of importance sampling and to appendix B.3 for details of efficient implementation.

---

**Algorithm 2** GreedySelection

1: **input**: a set of samples $\mathcal{D}$, $\bar{\mathcal{D}}_{\text{pool}}$, $\{\hat{\omega}_k, \hat{\omega}'_k\}_{k=1}^K$, threshold $\tau$, and $B$
2: $\mathcal{A}_0 = \emptyset$
3: **for** $b \in [B]$ **do**
4:     **for all** $x \in \mathcal{D} \backslash \mathcal{A}_{b-1}$ **do**
5:         $s_x \leftarrow \Delta_{\text{BALANCE}}(\mathcal{A}_{b-1} \bigcup \{x\})$
6:     $x_b \leftarrow \arg\max_{x \in \mathcal{D} \backslash \mathcal{A}_{b-1}} s_x$
7:     $\mathcal{A}_b \leftarrow \mathcal{A}_{b-1} \bigcup \{x_b\}$
8: **output**: batch $\mathcal{A}_B = \{x_1, \dots, x_B\}$

---

**Algorithm 3** BALANCE-Clustering

1: **input**: $\mathcal{C} \subset \mathcal{D}_{\text{pool}}, \bar{\mathcal{D}}_{\text{pool}}, \{\hat{\omega}_k, \hat{\omega}'_k\}_{k=1}^K$, threshold $\tau$, coldness parameter $\beta$, and cluster number $B$
2: sample initial centroids $\mathcal{O} = \{\mu_j\}_{j=1}^B \subset \mathcal{C}$ with $\text{p}(x) \sim \Delta_{\text{BALANCE}}(x)^\beta$
3: **while** $\mathcal{O}$ not converged **do**
4:     **for all** $x \in \mathcal{C}$ **do**
5:         $a_x \leftarrow \arg\max_j \text{I}_{\Delta_{\text{BALANCE}}}(x, \mu_j)$
6:     $\mathcal{S}_j \leftarrow \{x \in \mathcal{C} : a_x = j\}$
7:     **for all** $j \in [B]$ **do**
8:         $\mu_j \leftarrow \arg\max_{y \in \mathcal{S}_j} \sum_{x \in \mathcal{S}_j} \text{I}_{\Delta_{\text{BALANCE}}}(x, y)$
9: **output**: $\mathcal{S}_{1:B}, \mu_{1:B}$

---

## 3.3 STOCHASTIC BATCH SELECTION WITH POWER SAMPLING AND BALANCE-CLUSTERING

A simple approach to apply our new acquisition function to large batch is stochastic batch selection (Kirsch et al., 2021a), where we randomly select a batch with power distribution $\text{p}(x) \sim \Delta_{\text{BALANCE}}(x)^\beta$. We call this algorithm PowerBALANCE.

Next, we sought to further improve PowerBALANCE through a novel acquisition-function-driven clustering procedure. Inspired by Kothawade et al. (2021), we define a novel *information measure* $\text{I}_{\Delta_{\text{BALANCE}}}(x, y)$ for any two data samples $x$ and $y$ based on our acquisition function:

$$\text{I}_{\Delta_{\text{BALANCE}}}(x, y) = \Delta_{\text{BALANCE}}(x) + \Delta_{\text{BALANCE}}(y) - \Delta_{\text{BALANCE}}(\{x, y\}) \tag{3}$$

Intuitively, $I_{\Delta_{\text{BALANCE}}}(x, y)$ captures the amount of overlap between $x$ and $y$ w.r.t. $\Delta_{\text{BALANCE}}$. Therefore, it is natural to use it as a similarity measure for clustering, and use the cluster centroids as candidate queries. The BALANCE-Clustering algorithm is illustrated in algorithm 3.

Concretely, we first sample a subset $\mathcal{C} \subset \mathcal{D}_{\text{pool}}$ with $\text{p}(x) \sim \Delta_{\text{BALANCE}}(x)^{\beta}$ similar to (Kirsch et al., 2021a). The BALANCE-Clustering then runs an Lloyd's algorithm (with a non-Euclidean metric) to find $B$ cluster centroids (see Line 3-8 in algorithm 3): it takes the subset $\mathcal{C}, \{\hat{\omega}_k, \hat{\omega}'_k\}_{k=1}^{K}$, threshold $\tau$, coldness parameter $\beta$, and cluster number $B$ as input. It first samples initial centroids $\mathcal{O}$ with $\text{p}(x) \sim \Delta_{\text{BALANCE}}(x)^{\beta}$. Then, it iterates the process of adjusting the clusters and centroids until convergence and outputs $B$ cluster centroids as candidate queries.

## 4 EXPERIMENTS

In this section, we sought to show the efficacy of Batch-BALANCE on several diverse datasets, under both small batch setting and large batch setting. In the main paper, we focus on accuracy as the key performance metric as is commonly used in the literature; supplemental results with different evaluation metrics, including macro-average AUC, F1, and NLL, are provided in appendix D.

### 4.1 EXPERIMENTAL SETUP

**Datasets** In the main paper, we consider four datasets (i.e. MNIST (LeCun et al., 1998), Repeated-MNIST (Kirsch et al., 2019), Fashion-MNIST (Xiao et al., 2017) and EMNIST (Cohen et al., 2017)) as benchmarks for the small-batch setting, and two datasets (i.e. SVHN (Netzer et al., 2011), CIFAR (Krizhevsky et al., 2009)) as benchmarks for the large-batch setting. The reason for making the splits is that for the more challenging classification tasks on SVHN and CIFAR-10, the performance improvement for all baseline algorithms from a small batch (e.g., with batch size $< 50$) is hardly visible. We split each dataset into unlabeled AL pool $\mathcal{D}_{\text{pool}}$, initial training dataset $\mathcal{D}_{\text{train}}$, validation dataset $\mathcal{D}_{\text{val}}$, test dataset $\mathcal{D}_{\text{test}}$ and unlabeled dataset $\bar{\mathcal{D}}_{\text{pool}}$. $\bar{\mathcal{D}}_{\text{pool}}$ is only used for calculating the Hamming distance between hypotheses and is never used for training BNNs. For more experiment details about datasets, see appendix C.

**BNN models** At each AL iteration, we sample BNN posteriors given the acquired training dataset and select samples from $\mathcal{D}_{\text{pool}}$ to query labels according to the acquisition function of a chosen algorithm. To avoid overfitting, we train the BNNs with MC dropout at each iteration with early stopping. for MNIST, Repeated-MNIST, EMNIST, and FashionMNIST, we terminate the training of BNNs with patience of 3 epochs. For SVHN and CIFAR-10, we terminate the training of BNNs with patience of 20 epochs. The BNN with the highest validation accuracy is picked and used to calculate the acquisition functions. Additionally, we use weighted cross-entropy loss for training the BNN to mitigate the bias introduced by imbalanced training data. The BNN models are reinitialized in each AL iteration similar to Gal et al. (2017); Kirsch et al. (2019). It decorrelates subsequent acquisitions as the final model performance is dependent on a particular initialization. We use Adam optimizer (Kingma & Ba, 2017) for all the models in the experiments.

For cSG-MCMC, we use ResNet-18 (He et al., 2016) and run 400 epochs in each AL iteration. We set the number of cycles to 8 and initial step size to 0.5. 3 samples are collected in each cycle.

**Acquisition criterion for Batch-BALANCE under different bach sizes** For small AL batch with $B < 50$, Batch-BALANCE takes the greedy selection approach. For large AL batch with $B \geq 50$, BALANCE takes the clustering approach described in section 3.3. In the small batch-mode setting, if $b < 4$, Batch-BALANCE enumerates all $y_{1:b}$ configurations to compute the acquisition function $\Delta_{\text{(Batch-)BALANCE}}$ according to equation 2; otherwise, it uses $M = 10,000$ MC samples of $y_{1:b}$ and importance sampling to estimate $\Delta_{\text{Batch-BALANCE}}$ according to equation 6. All our results report the median of 6 trials, with lower and upper quartiles.

**Baselines** For the small-batch setting, we compare Batch-BALANCE with Random, Variation Ratio (Freeman & Freeman, 1965), Mean STD (Kendall et al., 2015) and BatchBALD. To the best of the authors' knowledge, Batch-BALD still achieves state-of-the-art performance for deep Bayesian AL with small batches. For large-batch setting, it is no longer feasible to run BatchBALD (Citovsky

et al., 2021); we consider other baseline models both in Bayesian setting, e.g., PowerBALD, and Non-Bayesian setting, e.g., CoreSet and BADGE.

## 4.2 COMPUTATIONAL COMPLEXITY ANALYSIS

Table 1 shows the computational complexity of the batch-mode AL algorithms evaluated in this paper. Here, $C$ denotes the number of classes, $B$ denotes the acquisition size, $K$ is the pair number of posterior samples and $M$ is the sample number for $y_{1:b}$ configurations. We assume the number of the hidden units is $H$. $T$ is # iterations for BALANCE-Clustering to converge and is usually less than 5. In figure 2 we plot the computation time for a single batch (in seconds) by different algorithms. As the batch size increases, variants of Batch-BALANCE (including Batch-BALANCE and PowerBALANCE as its special case) both outperforms CoreSet in run time. In later subsections, we will demonstrate that this gain in computational efficiency does not come at a cost of performance. We refer interested readers to section B.4 for extended discussion of computational complexity.

| AL algorithms | Complexity |
|---|---|
| Mean STD | $\mathcal{O}\left(\lvert\mathcal{D}_{\text{pool}}\rvert(CK + \log B)\right)$ |
| Variation Ratio | $\mathcal{O}\left(\lvert\mathcal{D}_{\text{pool}}\rvert(CK + \log B)\right)$ |
| PowerBALD | $\mathcal{O}\left(\lvert\mathcal{D}_{\text{pool}}\rvert(CK + \log B)\right)$ |
| BatchBALD | $\mathcal{O}\left(\lvert\mathcal{D}_{\text{pool}}\rvert BMK\right)$ |
| CoreSet (2-approx) | $\mathcal{O}(\lvert\mathcal{D}_{\text{pool}}\rvert HB)$ |
| BADGE | $\mathcal{O}(\lvert\mathcal{D}_{\text{pool}}\rvert HCB^2)$ |
| PowerBALANCE | $\mathcal{O}\left(\lvert\mathcal{D}_{\text{pool}}\rvert(C \cdot 2K + \log B)\right)$ |
| Batch-BALANCE (GreedySelection) | $\mathcal{O}\left(\lvert\mathcal{D}_{\text{pool}}\rvert BM \cdot 2K\right)$ |
| Batch-BALANCE (BALANCE-Clustering) | $\mathcal{O}(\lvert\mathcal{D}_{\text{pool}}\rvert C \cdot 2K + \lvert\mathcal{C}\rvert^2(C^2 \cdot 2K + T))$ |

Table 1: Computational complexity of AL algorithms.

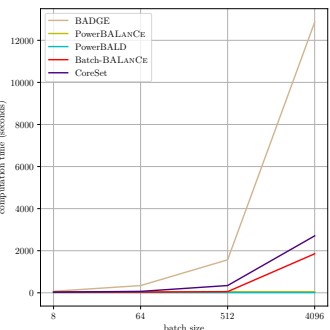

Figure 2: Run time vs. batch size.

## 4.3 BATCH-MODE DEEP BAYESIAN AL WITH SMALL BATCH SIZE

We compare 5 different models with acquisition sizes $B = 1$, $B = 3$ and $B = 10$ on MNIST dataset. $K = 100$ for all the methods. The threshold $\tau$ for Batch-BALANCE is annealed by setting $\tau$ to $\varepsilon/2$ in each AL loop. Note that when $B = 3$, we can compute the acquisition function with all $y_{1:b}$ configurations for $b = 1, 2, 3$. When $b \geq 4$, we approximate the acquisition function with importance sampling. Figure 3 (a)-(c) show that Batch-BALANCE are consistently better than other baseline methods for MNIST dataset.

We then compare Batch-BALANCE with other baseline methods on three datasets with balanced classes—Repeated-MNIST, Fashion-MNIST and EMNIST-Balanced. The acquisition size $B$ for Repeated-MNIST and Fashion-MNIST is 10 and is 5 for EMNIST-Balanced dataset. The threshold $\tau$ of Batch-BALANCE is annealed by setting $\tau = \varepsilon/4$[3]. The learning curves of accuracy are shown in figure 3 (d)-(f). For Repeated-MNIST dataset, BALD performs poorly and is worse than random selection. BatchBALD is able to cope with the replication after certain number of AL loops, which is aligned with result shown in Kirsch et al. (2019). Batch-BALANCE is able to beat all the other methods on this dataset. An ablation study about repetition number and performance can be found in appendix D.2. For Fashion-MNIST dataset, Batch-BALANCE outperforms random selection but the other methods fail. For EMNIST dataset, Batch-BALANCE is slightly better than BatchBALD.

We further compare different algorithms with two unbalanced datasets: EMNIST-ByMerge and EMNIST-ByClass. The $\tau$ for Batch-BALANCE is set $\varepsilon/4$ in each AL loop. $B = 5$ and $K = 10$ for all the methods. As pointed out by Kirsch et al. (2019), BatchBALD performs poorly in unbalanced dataset settings. BALANCE and Batch-BALANCE can cope with the unbalanced data settings. The result is shown in figure 3 (g) and (h). Further results on other datasets and under different metrics are provided in appendix D.

## 4.4 BATCH-MODE DEEP BAYESIAN AL WITH LARGE BATCH SIZE

**Batch-BALANCE with MC dropout** We test different AL models on two larger datasets with larger batch size. The acquisition batch size $B$ is set 1,000 and $\tau = \varepsilon/8$. We use VGG-11 as the BNN and train it on all the labeled data with patience equal to 20 epochs in each AL iteration. The

---

[3]Empirically we find that $\tau \in [\varepsilon/8, \varepsilon/2]$ works generally well for all datasets.

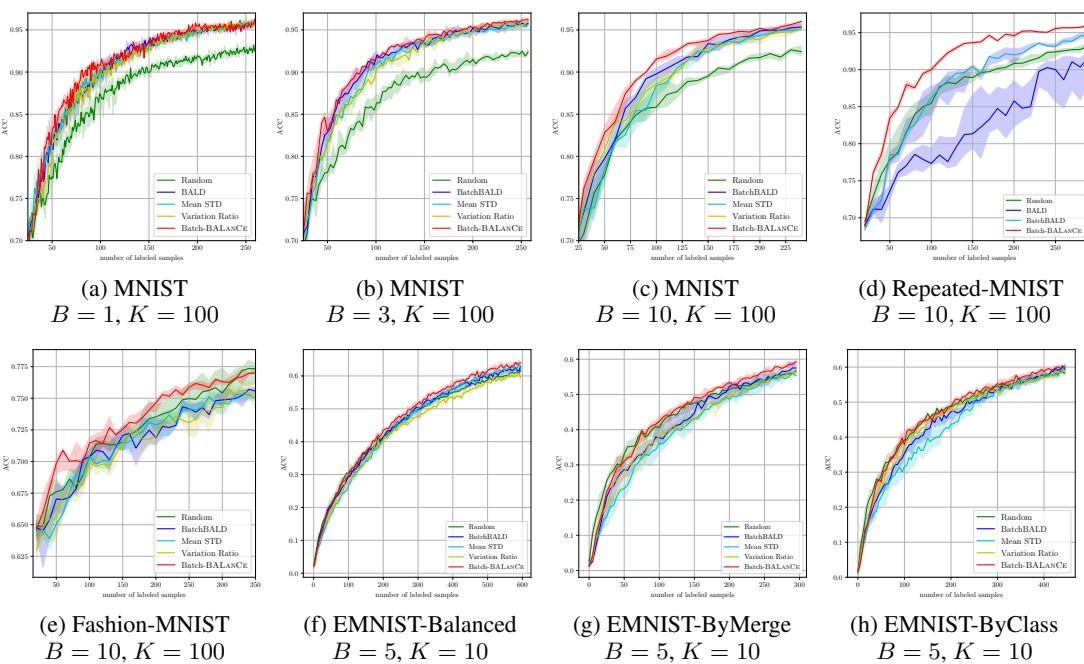

(a) MNIST
$B = 1, K = 100$

(b) MNIST
$B = 3, K = 100$

(c) MNIST
$B = 10, K = 100$

(d) Repeated-MNIST
$B = 10, K = 100$

(e) Fashion-MNIST
$B = 10, K = 100$

(f) EMNIST-Balanced
$B = 5, K = 10$

(g) EMNIST-ByMerge
$B = 5, K = 10$

(h) EMNIST-ByClass
$B = 5, K = 10$

Figure 3: Experimental results on MNIST, Repeated-MNIST, Fashion-MNIST, EMNIST-Balanced, EMNIST-ByClass and EMNIST-ByMerge datasets in the small-batch regime. For all plots, the $y$-axis represents accuracy and $x$-axis represents the number of queried examples.

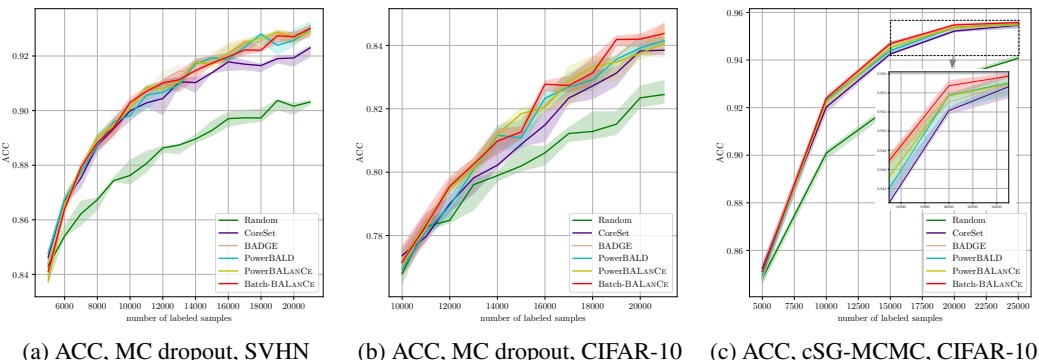

(a) ACC, MC dropout, SVHN

(b) ACC, MC dropout, CIFAR-10

(c) ACC, cSG-MCMC, CIFAR-10

Figure 4: Performance on SVHN and CIFAR-10 datasets in the large-batch regime.

VGG-11 is trained using SGD with fixed learning rate 0.001 and momentum 0.9. The size of $\mathcal{C}$ for Batch-BALANCE is set to $2B$. Similar to PowerBALD (Kirsch et al., 2021a), we also find that PowerBALanCe and BatchBALanCe are insensitive to $\beta$ and $\beta = 1$ works generally well. We thus set the coldness parameter $\beta = 1$ for all algorithms.

The performance of different AL models on these two datasets is shown in figure 4 (a) and (b). PowerBALD, PowerBALANCE, BADGE, and BatchBALANCE get similar performance on SVHN dataset. For CIFAR-10 dataset, BatchBALANCE shows compelling performance. Note that PowerBALANCE also performs well compared to other methods.

**Batch-BALANCE with cSG-MCMC** We test different AL models with cSG-MCMC on CIFAR-10. The acquisition batch size $B$ is 5,000. The size of $\mathcal{C}$ for Batch-BALANCE is set to $3B$. In order to apply CoreSet algorithm to BNN, we use the average activations of all posterior samples' final fully-connected layers as the representations. For BADGE, we use the label with maximum average predictive probability as the hallucinated label and use the average loss gradient of the last

layer induced by the hallucinated label as the representation. We can see from figure 4 (c) that Batch-BALanCe achieve the best performance.

## 5   RELATED WORK

**Pool-based batch-mode active learning**   Batch-mode AL has shown promising performance for practical AL tasks. Recent works, including both Bayesian (Houlsby et al., 2011; Gal et al., 2017; Kirsch et al., 2019) and non-Bayesian approaches (Sener & Savarese, 2017; Ash et al., 2019; Citovsky et al., 2021; Kothawade et al., 2021; Hacohen et al., 2022; Karanam et al., 2022), have been enormous and we hardly do it justice here. We mention what we believe are most relevant in the following. Among the Bayesian algorithms, Gal et al. (2017) choose a batch of samples with top acquisition functions. These methods can potentially suffer from choosing similar and redundant samples inside each batch. Kirsch et al. (2019) extended Houlsby et al. (2011) and proposed a batch-mode deep Bayesian AL algorithm, namely BatchBALD. Chen & Krause (2013) formalized a class of interactive optimization problems as adaptive submodular optimization problems and prove a greedy batch-mode approach to these problems is near-optimal as compared to the optimal batch selection policy. ELR (Roy & McCallum, 2001) focuses on a Bayesian estimate of the reduction in classification error and takes a one-step-look-ahead startegy. Inspired by ELR, WMOCU (Zhao et al., 2021) extends MOCU (Yoon et al., 2013) with a theoretical guarantee of convergence. However, none of these algorithms extend to the batch setting.

Among the non-Bayesian approaches, Sener & Savarese (2017) proposed a CoreSet approach to select a subset of representative points as a batch. BADGE (Ash et al., 2019) selects samples by using the k-MEAMS++ seeding algorithm on the $\mathcal{D}_{\text{pool}}$ representations, which are the gradient embeddings of DNN's last layer induced by hallucinated labels. Contemporary works propose AL algorithms that work for different settings including text classification (Tan et al., 2021), domain shift and outlier (Kirsch et al., 2021b), low-budget regime (Hacohen et al., 2022), very large batches (e.g., 100K or 1M) (Citovsky et al., 2021), rare classes and OOD data (Kothawade et al., 2021).

**Bayesian neural networks**   Bayesian methods have been shown to improve the generalization performance of DNNs (Hernández-Lobato & Adams, 2015; Blundell et al., 2015; Li et al., 2016b; Maddox et al., 2019), while providing principled representations of uncertainty. MCMC methods provides the gold standard of performance with smaller neural networks (Neal, 2012). SG-MCMC methods (Welling & Teh, 2011; Chen et al., 2014; Ding et al., 2014; Li et al., 2016a) provide a promising direction for sampling-based approaches in Bayesian deep learning. cSG-MCMC (Zhang et al., 2019) proposes a cyclical stepsize schedule, which indeed generates samples with high fidelity to the true posterior (Fortuin et al., 2021; Izmailov et al., 2021). Another BNN posterior approximation is MC dropout (Gal & Ghahramani, 2016; Kingma et al., 2015). We investigate both the cSG-MCMC and MC dropout methods as representative BNN models in our empirical study.

**Semi-supervised learning**   Semi-supervised learning leverages both unlabeled and labeled examples in the training process (Kingma et al., 2014; Rasmus et al., 2015). Some work has combined AL and semi-supervised learning (Wang et al., 2016; Sener & Savarese, 2017; Sinha et al., 2019). Our methods are different from these methods since our methods never leverage unlabeled data to train the models, but rather use the unlabeled pool to inform the selection of data points for AL.

## 6   CONCLUSION AND DISCUSSION

We have proposed a scalable batch-mode deep Bayesian active learning framework, which leverages the hypothesis structure captured by equivalence classes without explicitly constructing them. Batch-BALanCe selects a batch of samples at each iteration which can reduce the overhead of retraining the model and save labeling effort. By combining insights from decision-theoretic active learning and diversity sampling, the proposed algorithms achieve compelling performance efficiently on active learning benchmarks both in small batch- and large batch-mode settings. Given the promising empirical results on the standard benchmark datasets explored in this paper, we are further interested in understanding the theoretical properties of the equivalence annealing algorithm under controlled studies as future work.

**Acknowledgement.** This work was supported in part by C3.ai DTI Research Award 049755, NSF award 2037026 and an NVIDIA GPU grant. Any opinions, findings, conclusions, or recommendations expressed in this material are those of the authors and do not necessarily reflect the views of any funding agencies.

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

CONTENTS

# Appendices

## Table of Contents

# A  PRELIMINARY WORKS

## A.1  THE MOST INFORMATIVE SELECTION CRITERION

BALD uses mutual information between the model prediction for each sample and parameters of the model as the acquisition function. It captures the reduction of model uncertainty by receiving a label $y$ of a data point $x$: $\mathbb{I}(y; \omega \mid x, \mathcal{D}_{\text{train}}) = \mathbb{H}(y \mid x, \mathcal{D}_{\text{train}}) - \mathbb{E}_{\text{p}(\omega \mid \mathcal{D}_{\text{train}})}[\mathbb{H}(y \mid x, \omega, \mathcal{D}_{\text{train}})]$ where $\mathbb{H}$ denotes the Shannon entropy (Shannon, 1948). Kirsch et al. (2019) further proposed BatchBALD as an extension of BALD whereby the mutual information between a joint of multiple data points and the model parameters is estimated as

$$\Delta_{\text{BatchBALD}}(x_{1:b} \mid \mathcal{D}_{\text{train}}) \triangleq \mathbb{I}(y_{1:b}; \omega \mid x_{1:b}, \mathcal{D}_{\text{train}}).$$

**Limitation of the BALD algorithm**    BALD can be ineffective when the hypothesis samples are heavily biased and cluttered towards sub-optimal hypotheses. Below, we provide a concrete example where such selection criterion may be undesirable.

Figure 5: A stylized example where the most informative selection criterion underperforms the equivalence-class-based criterion.

Consider the problem shown in figure 5. The hypothesis class $\mathcal{H} = \{h_1, \ldots, h_n\}$ is structured such that

$$d_{\text{H}}(h_i, h_j) = \begin{cases} 2^{1-i} - 2^{1-j} & \text{if } i < j, \\ 2^{1-j} - 2^{1-i} & \text{o.w.} \end{cases}$$

where $d_{\text{H}}(h_i, h_j)$ denotes the fraction of labels $h_i$ and $h_j$ disagree upon when making predictions on $i.i.d.$ samples of data points. We further assume that for any subset of hypotheses $S \subseteq \mathcal{H}$, there exists a data point whose label they agree upon. Assume each hypothesis $h_i$ has an equal probability and the target error rate is $\sigma$. On the one hand, note that BALD does not consider $d_{\text{H}}(h_i, h_j)$, and therefore on average it requires $\log n$ examples to identify any target hypothesis. On the other hand, to achieve a target error rate of $\sigma$, one only needs to differentiate all pairs of hypotheses $h_i, h_j$ of distance $d_{\text{H}}(h_i, h_j) > \sigma$ (i.e., by selecting training examples to rule out at least one of $h_i$, $h_j$). Therefore, a "smarter" AL policy could query examples to sequentially check the consistency of $h_1, h_2, \ldots, h_n$ until all remaining hypotheses are within distance $\sigma$. It is easy to check that this requires $\log(1/\sigma)$ examples before reaching the error rate $\sigma$. The gap between BALD and the above policy $\frac{\log n}{\log(1/\sigma)}$ could be large as $n$ increases.

## A.2  EQUIVALENCE CLASS EDGE CUTTING

Consider the problem statement in section 2.1. If $\sigma = 0$ and tests are noise-free, this problem can be solved near-optimally by the *equivalence class edge cutting* (EC$^2$) algorithm (Golovin et al., 2010). EC$^2$ employs an edge-cutting strategy based on a weighted graph $G = (\mathcal{H}, \mathcal{E})$, where vertices represent hypotheses and edges link hypotheses that we want to distinguish between. Here $\mathcal{E} \triangleq \{\{h, h'\} : r(h) \neq r(h')\}$ contains all pairs of hypotheses that have different equivalence classes. We define a weight function $W : \mathcal{E} \to \mathbb{R}_{\geq 0}$ by $W(\{h, h'\}) \triangleq \text{p}(h) \cdot \text{p}(h')$. A sample $x$ with label $y$ is said to "cut" an edge, if at least one hypothesis is inconsistent with $y$. Denote $\mathcal{E}(x, y) \triangleq \{\{h, h'\} \in \mathcal{E} : \text{p}(y \mid x, h) = 0 \vee \text{p}(y \mid x, h') = 0\}$ as the set of edges cut by labeling $x$ as $y$. The EC$^2$ objective is then defined as the total weight of edges cut by the current $\mathcal{D}_{\text{train}}$: $f_{\text{EC}^2}(\mathcal{D}_{\text{train}}) \triangleq W\left(\bigcup_{(x,y) \in \mathcal{D}_{\text{train}}} \mathcal{E}(x, y)\right)$. EC$^2$ algorithm greedily maximizes this objective per iteration. The acquisition function for EC$^2$ is

$$\Delta_{\text{EC}^2}(x \mid \mathcal{D}_{\text{train}}) \triangleq \mathbb{E}_y\left[f(\mathcal{D}_{\text{train}} \cup \{(x, y)\}) - f(\mathcal{D}_{\text{train}}) \mid \mathcal{D}_{\text{train}}\right]. \tag{4}$$

### A.3 THE EQUIVALENCE CLASS EDGE DISCOUNTING ALGORITHM

In the noisy setting, the acquisition function of *Equivalence Class Edge Discounting* algorithm (ECED) (Chen et al., 2016) takes undesired contribution by noise into account. Given a data point and its label $(x, y)$, ECED discounts all model parameters by their likelihood ratio: $\lambda_{h,y} \triangleq \frac{\mathrm{p}(y|h,x)}{\max_{y'} \mathrm{p}(y'|h,x)}$. After we get $\mathcal{D}_{\mathrm{train}}$, the value of assigning label $y$ to a data point $x$ is defined as the total amount of edge weight discounted: $\delta(x, y \mid \mathcal{D}_{\mathrm{train}}) \triangleq \sum_{\{h,h'\} \in \mathcal{E}} \mathrm{p}(h, \mathcal{D}_{\mathrm{train}}) \mathrm{p}(h', \mathcal{D}_{\mathrm{train}}) \cdot (1 - \lambda_{h,y} \lambda_{h',y})$, where $\mathcal{E} = \{\{h, h'\} : r(h) \neq r(h')\}$ consists of all unordered pairs of hypothesis corresponding to different equivalence classes. Further, ECED augments the above value function $\delta$ with an offset value such that the value of a non-informative test is 0. The offset value of labeling $x$ as label $y$ is defined as: $\nu(x, y \mid \mathcal{D}_{\mathrm{train}}) \triangleq \sum_{\{h,h'\} \in \mathcal{E}} \mathrm{p}(h, \mathcal{D}_{\mathrm{train}}) \mathrm{p}(h', \mathcal{D}_{\mathrm{train}}) \cdot (1 - \max_h \lambda_{h,y}^2)$. The overall acquisition function of ECED is:

$$\Delta_{\mathrm{ECED}}(x \mid \mathcal{D}_{\mathrm{train}}) \triangleq \mathbb{E}_y \left[ \delta(x, y \mid \mathcal{D}_{\mathrm{train}}) - \nu(x, y \mid \mathcal{D}_{\mathrm{train}}) \right]. \tag{5}$$

## B ALGORITHMIC DETAILS

### B.1 DERIVATION OF ACQUISITION FUNCTIONS OF BALANCE AND BATCH-BALANCE

In each AL loop, the ECED algorithm selects a sample from AL pool according to the acquisition function

$$\Delta_{\mathrm{ECED}}(x \mid \mathcal{D}_{\mathrm{train}}) \triangleq \mathbb{E}_y \left[ \sum_{\{\omega,\omega'\} \in \mathcal{E}} W_{\omega,\omega'} \left( 1 - \lambda_{\omega,y} \lambda_{\omega',y} - \left( 1 - \max_{\omega} \lambda_{\omega,y}^2 \right) \right) \right],$$

where $\mathcal{E}$ is the total edges with adjacent nodes in different equivalence classes and $\lambda_{\omega,y} = \frac{\mathrm{p}(y|\omega)}{\max_{y'} \mathrm{p}(y'|\omega)}$. $W_{\omega,\omega'}$ is the weight for edge $\{\omega, \omega'\}$ which is maintained by ECED algorithm. After we observe $y$ of selected $x$, we update the weights of all edges with $W_{\omega,\omega'} = W_{\omega,\omega'} \cdot \mathrm{p}(y \mid \omega) \mathrm{p}(y \mid \omega')$. In the deep Bayesian AL setting, the offset term $1 - \max_{\omega} \lambda_{\omega,y}^2$ can be removed when we use deep BNN. However, we can not enumerate all $\{\omega, \omega'\} \in \mathcal{E}$ in this setting since there are an infinite number of hypotheses in the hypothesis space. Moreover, we can not even estimate the acquisition function of ECED on a subset of sampled hypotheses by MC dropouts since building equivalence classes with best $\epsilon$ is NP-hard.

If we sample $\{\omega, \omega'\}$ according to posterior $\mathrm{p}(\omega \mid \mathcal{D}_{\mathrm{train}})$ and check whether $\{\omega, \omega'\} \in \hat{\mathcal{E}}$ by Hamming distance in the way we describe in section 3.1, we will get

$$\Delta_{\mathrm{ECED}}(x \mid \mathcal{D}_{\mathrm{train}})$$

$$\approx \mathbb{E}_y \left[ \sum_{\{\omega,\omega'\} \in \mathcal{E}} W_{\omega,\omega'} \left( 1 - \lambda_{\omega,y} \lambda_{\omega',y} \right) \right]$$

$$\approx \mathbb{E}_y \left[ \mathbb{E}_{\omega,\omega' \sim \mathrm{p}(\omega | \mathcal{D}_{\mathrm{train}})} \mathbb{1}_{d_{\mathrm{H}}(\omega,\omega') > \tau} \cdot \frac{W_{\omega,\omega'}}{\mathrm{p}(\omega \mid \mathcal{D}_{\mathrm{train}}) \mathrm{p}(\omega' \mid \mathcal{D}_{\mathrm{train}})} \cdot (1 - \lambda_{\omega,y} \lambda_{\omega',y}) \right]$$

$$\propto \mathbb{E}_y \left[ \mathbb{E}_{\omega,\omega' \sim \mathrm{p}(\omega | \mathcal{D}_{\mathrm{train}})} \mathbb{1}_{d_{\mathrm{H}}(\omega,\omega') > \tau} \cdot (1 - \lambda_{\omega,y} \lambda_{\omega',y}) \right].$$

Inspired by the weight discounting mechanism of ECED, we define the acquisition function of BALANCE $\Delta_{\mathrm{BALANCE}}(x \mid \mathcal{D}_{\mathrm{train}})$ as

$$\Delta_{\mathrm{BALANCE}}(x \mid \mathcal{D}_{\mathrm{train}}) \triangleq \mathbb{E}_y \left[ \mathbb{E}_{\omega,\omega' \sim \mathrm{p}(\omega | \mathcal{D}_{\mathrm{train}})} \mathbb{1}_{d_{\mathrm{H}}(\omega,\omega') > \tau} \cdot (1 - \lambda_{\omega,y} \lambda_{\omega',y}) \right].$$

After we get $K$ pairs of MC dropouts, the acquisition function $\Delta_{\mathrm{BALANCE}}(x \mid \mathcal{D}_{\mathrm{train}})$ can be approximated as follows:

$$\Delta_{\mathrm{BALANCE}}(x \mid \mathcal{D}_{\mathrm{train}})$$

$$= \mathbb{E}_{\mathrm{p}(\omega | \mathcal{D}_{\mathrm{train}})} \mathbb{E}_{\mathrm{p}(y|\omega)} \left[ \mathbb{E}_{\omega,\omega' \sim \mathrm{p}(\omega | \mathcal{D}_{\mathrm{train}})} \mathbb{1}_{d_{\mathrm{H}}(\omega,\omega') > \tau} \left( 1 - \lambda_{\omega,y} \lambda_{\omega',y} \right) \right]$$

$$\approx \sum_{\hat{y}} \left( \frac{1}{2K} \sum_{k=1}^{K} \mathrm{p}(\hat{y} \mid \hat{\omega}_k) + \mathrm{p}(\hat{y} \mid \hat{\omega}'_k) \right) \left[ \frac{1}{K} \sum_{k=1}^{K} \mathbb{1}_{d_{\mathrm{H}}(\hat{\omega}_k, \hat{\omega}'_k) > \tau} \left( 1 - \lambda_{\hat{\omega}_k, \hat{y}} \lambda_{\hat{\omega}'_k, \hat{y}} \right) \right].$$

In batch-mode setting, the acquisition function of Batch-BALANCE for a batch $x_{1:b}$ is

$$\Delta_{\text{Batch}-\text{BALANCE}}(x_{1:b} \mid \mathcal{D}_{\text{train}}) \triangleq \mathbb{E}_{y_{1:b}} \left[ \mathbb{E}_{\omega,\omega' \sim \text{p}(\omega \mid \mathcal{D}_{\text{train}})} \mathbb{1}_{d_{\text{H}}(\omega,\omega')>\tau} \cdot (1 - \lambda_{\omega,y_{1:b}} \lambda_{\omega',y_{1:b}}) \right].$$

Similar to the fully sequential setting, we can approximate $\Delta_{\text{Batch}-\text{BALANCE}}(x_{1:b} \mid \mathcal{D}_{\text{train}})$ with $K$ pairs of MC dropouts. The $x_{1:b}$ are chosen in a greedy manner. For iteration $b$ inside a batch, the $x_{1:b-1}$ are fixed and $x_b$ is selected according to

$$\Delta_{\text{Batch}-\text{BALANCE}}(x_{1:b} \mid \mathcal{D}_{\text{train}})$$
$$=\mathbb{E}_{\text{p}(\omega \mid \mathcal{D}_{\text{train}})} \mathbb{E}_{\text{p}(y_{1:b} \mid \omega)} \left[ \mathbb{E}_{\omega,\omega' \sim \text{p}(\omega \mid \mathcal{D}_{\text{train}})} \mathbb{1}_{d_{\text{H}}(\omega,\omega')>\tau} (1 - \lambda_{\omega,y_{1:b}} \lambda_{\omega',y_{1:b}}) \right]$$
$$\approx \sum_{\hat{y}_{1:b}} \left( \frac{1}{2K} \sum_{k=1}^{K} \text{p}(\hat{y}_{1:b} \mid \hat{\omega}_k) + \text{p}(\hat{y}_{1:b} \mid \hat{\omega}'_k) \right) \left[ \frac{1}{K} \sum_{k=1}^{K} \mathbb{1}_{d_{\text{H}}(\hat{\omega}_k,\hat{\omega}'_k)>\tau} \left( 1 - \lambda_{\hat{\omega}_k,\hat{y}_{1:b}} \lambda_{\hat{\omega}'_k,\hat{y}_{1:b}} \right) \right].$$

### B.2 Importance sampling of configurations

When $b$ becomes large, it is infeasible to enumerate all label configurations $y_{1:b}$. We use $M$ MC samples of $y_{1:b}$ to estimate the acquisition function and importance sampling to further reduce the computational time[4]. Given that $\text{p}(y_{1:b} \mid \omega)$ can be factorized as $\text{p}(y_{1:b-1} \mid \omega) \cdot \text{p}(y_b \mid \omega)$, the acquisition function can be written as:

$$\Delta_{\text{Batch}-\text{BALANCE}}(x_{1:b} \mid \mathcal{D}_{\text{train}})$$
$$\triangleq \mathbb{E}_{y_{1:b}} \left[ \mathbb{E}_{\text{p}(\omega \mid \mathcal{D}_{\text{train}})} \mathbb{1}_{d_{\text{H}}(\omega_k,\omega'_k)>\tau} (1 - \lambda_{\omega,y_{1:b}} \lambda_{\omega',y_{1:b}}) \right]$$
$$= \mathbb{E}_{\text{p}(\omega \mid \mathcal{D}_{\text{train}})} \mathbb{E}_{\text{p}(y_{1:b} \mid \omega)} \left[ \mathbb{E}_{\omega,\omega' \sim \text{p}(\omega \mid \mathcal{D}_{\text{train}})} \mathbb{1}_{d_{\text{H}}(\omega_k,\omega'_k)>\tau} (1 - \lambda_{\omega,y_{1:b}} \lambda_{\omega',y_{1:b}}) \right]$$
$$= \mathbb{E}_{\text{p}(\omega \mid \mathcal{D}_{\text{train}})} \mathbb{E}_{\text{p}(y_{1:b-1} \mid \omega)} \mathbb{E}_{\text{p}(y_b \mid \omega)} \left[ \mathbb{E}_{\omega,\omega' \sim \text{p}(\omega \mid \mathcal{D}_{\text{train}})} \mathbb{1}_{d_{\text{H}}(\omega_k,\omega'_k)>\tau} (1 - \lambda_{\omega,y_{1:b}} \lambda_{\omega',y_{1:b}}) \right]$$

Suppose we have $M$ samples of $y_{1:b-1}$ from $\text{p}(y_{1:b-1})$, we perform importance sampling using $\text{p}(y_{1:b-1})$ to estimate the acquisition function:

$$\Delta_{\text{Batch}-\text{BALANCE}}(x_{1:b} \mid \mathcal{D}_{\text{train}})$$
$$= \mathbb{E}_{\text{p}(\omega \mid \mathcal{D}_{\text{train}})} \mathbb{E}_{\text{p}(y_{1:b-1})} \frac{\text{p}(y_{1:b-1} \mid \omega)}{\text{p}(y_{1:b-1})} \mathbb{E}_{\text{p}(y_b \mid \omega)} \left[ \mathbb{E}_{\omega,\omega' \sim \text{p}(\omega \mid \mathcal{D}_{\text{train}})} \mathbb{1}_{d_{\text{H}}(\omega,\omega')>\tau} (1 - \lambda_{\omega,y_{1:b}} \lambda_{\omega',y_{1:b}}) \right]$$
$$= \mathbb{E}_{\text{p}(y_{1:b-1})} \mathbb{E}_{\text{p}(\omega \mid \mathcal{D}_{\text{train}})} \mathbb{E}_{\text{p}(y_b \mid \omega)} \frac{\text{p}(y_{1:b-1} \mid \omega)}{\text{p}(y_{1:b-1})} \left[ \mathbb{E}_{\omega,\omega' \sim \text{p}(\omega \mid \mathcal{D}_{\text{train}})} \mathbb{1}_{d_{\text{H}}(\omega,\omega')>\tau} (1 - \lambda_{\omega,y_{1:b}} \lambda_{\omega',y_{1:b}}) \right]$$
$$\approx \frac{1}{M} \sum_{\hat{y}_{1:b-1}}^{M} \sum_{\hat{y}_b} \frac{\frac{1}{K} \sum_{k=1}^{K} \text{p}(\hat{y}_{1:b-1} \mid \hat{\omega}_k) \text{p}(\hat{y}_b \mid \hat{\omega}_k) + \text{p}(\hat{y}_{1:b-1} \mid \hat{\omega}'_k) \text{p}(\hat{y}_b \mid \hat{\omega}'_k)}{\text{p}(\hat{y}_{1:b-1})} \cdot$$
$$\left[ \frac{1}{K} \sum_{k=1}^{K} \mathbb{1}_{d_{\text{H}}(\hat{\omega}_k,\hat{\omega}'_k)>\tau} \left( 1 - \lambda_{\hat{\omega}_k,\hat{y}_{1:b}} \lambda_{\hat{\omega}'_k,\hat{y}_{1:b}} \right) \right]$$
$$= \left( \frac{1}{K} \mathbb{1}_{d_{\text{H}}(\hat{\omega}_k,\hat{\omega}'_k)>\tau} \right)^{\top} \left( 1 - \frac{\hat{P}_{1:b-1} \otimes \hat{P}_b}{\hat{A}_{1:b}} \odot \frac{\hat{P}'_{1:b-1} \otimes \hat{P}'_b}{\hat{A}'_{1:b}} \right) \left( \frac{1}{M} \frac{\hat{P}_{1:b-1}^{\top} \hat{P}_b + \hat{P}'^{\top}_{1:b-1} \hat{P}'_b}{\mathbb{1}^{\top} \left( \hat{P}_{1:b-1} + \hat{P}'_{1:b-1} \right)} \right)^{\top}. \tag{6}$$

Here we save $\text{p}(\hat{y}_{1:b-1} \mid \hat{\omega}_k)$ and $\text{p}(\hat{y}_{1:b-1} \mid \hat{\omega}'_k)$ for $M$ samples in $\hat{P}_{1:b-1}$ and $\hat{P}'_{1:b-1}$. The shape of $\hat{P}_{1:b-1}$ and $\hat{P}'_{1:b-1}$ is $K \times M$. $\odot$ is element-wise matrix multiplication and $\otimes$ is the outer-product operator along first dimension. After the outer product operation, we can reshape the matrix by flattening all the dimensions after the 1st dimension. $\mathbb{1}$ is a matrix of 1s with shape $K \times 1$. $\hat{P}_{1:b-1}^{\top} \hat{P}_b$ and $\hat{P}'^{\top}_{1:b-1} \hat{P}'_b$ are of shape $M \times C$ and their sum is reshape to $1 \times MC$ after divided by $\mathbb{1}^{\top} \left( \hat{P}_{1:b-1} + \hat{P}'_{1:b-1} \right)$.

---

[4]A similar importance sampling procedure was proposed in Kirsch et al. (2019) to estimate the mutual information. Here, we show how one can adapt the strategy to enable efficient estimation of $\Delta_{\text{Batch}-\text{BALANCE}}$.

### B.3 Efficient implementation for greedy selection

In algorithm 2, we can store $\mathrm{p}(\hat{y}_{1:b-1} \mid \hat{\omega}_k)$ in a matrix $\hat{P}_{1:b-1}$ and $\mathrm{p}(\hat{y}_{1:b-1} \mid \hat{\omega}'_k)$ in matrix $\hat{P}_{1:b-1}$ for iteration $b-1$. The shape of $\hat{P}_{1:b-1}$ and $\hat{P}'_{1:b-1}$ is $K \times C^{b-1}$. $\mathrm{p}(\hat{y}_b \mid \hat{\omega}_k)$ can be stored in $\hat{P}_b$ and $\mathrm{p}(\hat{y}_b \mid \hat{\omega}'_k)$ in $\hat{P}'_b$. The shape of $\hat{P}_b$ and $\hat{P}'_b$ is $K \times C$. Then, we compute probability of $\mathrm{p}(\hat{y}_{1:b})$ as follows:

$$
\begin{aligned}
\mathrm{p}(\hat{y}_{1:b}) =& \frac{1}{2K} \sum_{k=1}^{K} \mathrm{p}(\hat{y}_{1:b} \mid \hat{\omega}_k) + \mathrm{p}(\hat{y}_{1:b} \mid \hat{\omega}'_k) \\
=& \frac{1}{2K} \sum_{k=1}^{K} \mathrm{p}(\hat{y}_{1:b-1} \mid \hat{\omega}_k)\mathrm{p}(\hat{y}_b \mid \hat{\omega}_k) + \mathrm{p}(\hat{y}_{1:b-1} \mid \hat{\omega}'_k)\mathrm{p}(\hat{y}_b \mid \hat{\omega}'_k) \\
=& \frac{1}{2K} (\hat{P}_{1:b-1}^{\top}\hat{P}_b + \hat{P}_{1:b-1}^{\prime\top}\hat{P}'_b).
\end{aligned}
$$

The $\hat{P}_{1:b-1}^{\top}\hat{P}_b$ and $\hat{P}_{1:b-1}^{\prime\top}\hat{P}'_b$ can be flattened to shape $1 \times C^b$ after matrix multiplication. We store $\max_{\hat{y}_{1:b-1}} \mathrm{p}(\hat{y}_{1:b-1} \mid \hat{\omega}_k)$ in a matrix $\hat{A}_{1:b-1}$ and $\max_{\hat{y}'_{1:b-1}} \mathrm{p}(\hat{y}'_{1:b-1} \mid \hat{\omega}'_k)$ in a matrix $\hat{A}'_{1:b-1}$. The shape of $\hat{A}_{1:b-1}$ and $\hat{A}'_{1:b-1}$ is $K \times 1$. We can compute $\lambda_{\hat{\omega},\hat{y}_{1:b}}$ inside edge weight discount expression by

$$
\begin{aligned}
\hat{A}_{1:b} &= \hat{A}_{1:b-1} \odot \max_{\hat{y}_b} \hat{P}_b; \\
\mathrm{p}(\hat{y}_{1:b} \mid \hat{\omega}_k) &= \mathrm{p}(\hat{y}_{1:b-1} \mid \hat{\omega}_k)\mathrm{p}(\hat{y}_b \mid \hat{\omega}_k) = \hat{P}_{1:b-1} \otimes \hat{P}_b; \\
\lambda_{\hat{\omega},\hat{y}_{1:b}} &= \frac{\mathrm{p}(\hat{y}_{1:b} \mid \hat{\omega}_k)}{\max_{\hat{y}_{1:b}} \mathrm{p}(\hat{y}_{1:b} \mid \hat{\omega}_k)} = \frac{\hat{P}_{1:b-1} \otimes \hat{P}_b}{\hat{A}_{1:b}}.
\end{aligned}
$$

$\odot$ is element-wise matrix multiplication and $\otimes$ is the outer-product operator along the first dimension. After the outer product operation, we can reshape the matrix by flattening all the dimensions after 1st dimension to maintain consistency. Similarly, we can compute $\hat{A}'_{1:b}$, $\mathrm{p}(\hat{y}_{1:b} \mid \hat{\omega}'_k)$ and $\lambda_{\hat{\omega}',\hat{y}_{1:b}}$ with matrix operations. The indicator function $\mathbb{1}_{d_{\mathrm{H}}(\hat{\omega}_k,\hat{\omega}'_k)>\tau}$ can be stored in a matrix with shape $K \times 1$. The acquisition function can be computed with all matrix operations as follows:

$$
\begin{aligned}
&\Delta_{\mathrm{Batch-BALANCE}}(x_{1:b} \mid \mathcal{D}_{\mathrm{train}}) \\
=& \mathbb{E}_{\mathrm{p}(\omega|\mathcal{D}_{\mathrm{train}})} \mathbb{E}_{\mathrm{p}(y_{1:b}|\omega)} \left[ \mathbb{E}_{\omega,\omega' \sim \mathrm{p}(\omega|\mathcal{D}_{\mathrm{train}})} \mathbb{1}_{d_{\mathrm{H}}(\omega,\omega')>\tau} \left( 1 - \lambda_{\omega,y_{1:b}}\lambda_{\omega',y_{1:b}} \right) \right] \\
\approx& \sum_{\hat{y}_{1:b}} \left( \frac{1}{2K} \sum_{k=1}^{K} \mathrm{p}(\hat{y}_{1:b} \mid \hat{\omega}_k) + \mathrm{p}(\hat{y}_{1:b} \mid \hat{\omega}'_k) \right) \left[ \frac{1}{K} \sum_{k=1}^{K} \mathbb{1}_{d_{\mathrm{H}}(\hat{\omega}_k,\hat{\omega}'_k)>\tau} \left( 1 - \lambda_{\hat{\omega}_k,\hat{y}_{1:b}}\lambda_{\hat{\omega}'_k,\hat{y}_{1:b}} \right) \right] \\
=& \left( \frac{1}{K} \mathbb{1}_{D(\hat{\omega}_k,\hat{\omega}'_k)>\tau} \right)^{\top} \left( 1 - \frac{\hat{P}_{1:b-1} \otimes \hat{P}_b}{\hat{A}_{1:b}} \odot \frac{\hat{P}'_{1:b-1} \otimes \hat{P}'_b}{\hat{A}'_{1:b}} \right) \left[ \frac{1}{2K} (\hat{P}_{1:b-1}^{\top}\hat{P}_b + \hat{P}_{1:b-1}^{\prime\top}\hat{P}'_b) \right]^{\top}.
\end{aligned}
$$

### B.4 Detailed computational complexity discussion

As demonstrated in figure 2, figure 6, and table 1, the computational complexity of our algorithm PowerBALANCE shares is comparable to PowerBALD. They all need to estimate the acquisition function value for each data point in the AL pool and then choose the top $B$ data points after adding Gumbel-distributed noise to the log values. However, the power sampling-based methods have limited performance due to the lack of interaction between selected samples and non-selected samples during sampling. We can further improve the performance of PowerBALANCE with Batch-BALANCE. The computation complexity of Batch-BALANCE for large batch setting are proportional to $B^2$ when downsampled with subset size $|\mathcal{C}| = cB$ and $c$ is a small constant. Its computational complexity is similar to that of BADGE and CoreSet.

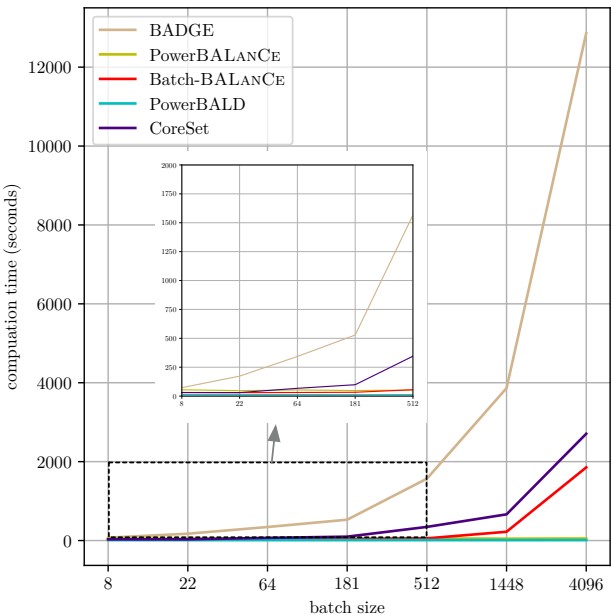

Figure 6: Computation time (in seconds) vs. batch size for different AL algorithms

## C EXPERIMENTAL SETUP: DATASETS AND IMPLEMENTATION DETAILS

### C.1 DATASETS USED IN THE MAIN PAPER

*MNIST.* We randomly split MNIST training dataset into $\mathcal{D}_{\mathrm{val}}$ with 10,000 samples, $\bar{\mathcal{D}}_{\mathrm{pool}}$ with 10,000 samples and $\mathcal{D}_{\mathrm{pool}}$ with the rest. The initial training dataset contains 20 samples with 2 samples in each class chosen from the AL pool. The BNN model architecture is similar to Kirsch et al. (2019). It consists of two blocks of [convolution, dropout, max-pooling, relu] followed by a two-layer MLP that a two-layer MLP and one dropout between the two layers. The dropout probability is 0.5 in the dropout layers.

*Repeated-MNIST.* Kirsch et al. (2019) show that applying BALD to a dataset that contains many (near) replicated data points leads to poor performance. We again randomly split the MNIST training dataset similar to the settings used on MNIST dataset. We replicate all the samples in AL pool two times and add isotropic Gaussian noise with a standard deviation of 0.1 after normalizing the dataset. The BNN architecture is the same as the one used on MNIST dataset.

*EMNIST.* We further consider the EMNIST dataset under 3 different settings: EMNIST-Balanced, EMNIST-ByClass, and EMNIST-ByMerge. The EMNIST-Balanced contains 47 classes with balanced digits and letters. EMNIST-ByMerge includes digits and letters for a total of 47 unbalanced classes. EMNIST-ByClass represents the most useful organization for classification as it contains the segmented digits and characters for 62 classes comprising [0-9],[a-z], and [A-Z]. We randomly split the training set into $\mathcal{D}_{\mathrm{val}}$ with 18,800 images, $\bar{\mathcal{D}}_{\mathrm{pool}}$ with 18,800 images and $\mathcal{D}_{\mathrm{pool}}$ with the rest of the samples. Similar to Kirsch et al. (2019), we do not use an initial dataset and instead perform the initial acquisition step with the randomly initialized model. The model architecture contains three blocks of [convolution, dropout, max-pooling, relu], with 32, 64, and 128 3x3 convolution filters and 2x2 max pooling. We add a two-layer MLP following the three blocks. 4 dropout layers in total are in each block and MLP with dropout probability 0.5.

*Fashion-MNIST.* Fashion-MNIST is a dataset of Zalando's article images that consists of a training set of 60,000 examples and a test set of 10,000 examples. Each example is a 28x28 grayscale image, associated with a label from 10 classes. We randomly split Fashion-MNIST training dataset into $\mathcal{D}_{\mathrm{val}}$ with 10,000 samples, $\bar{\mathcal{D}}_{\mathrm{pool}}$ with 10,000 samples, and $\mathcal{D}_{\mathrm{pool}}$ with the rest of samples. We obtain the initial training dataset that contains 20 samples with 2 samples in each class randomly

chosen from the AL pool. The model architecture is similar to the one used on EMNIST dataset with 10 units in the last MLP.

*SVHN.* We randomly select initial training dataset with 5,000 samples, $\bar{\mathcal{D}}_{\text{pool}}$ with 2,000 samples, and validation dataset $\mathcal{D}_{\text{val}}$ with 5,000 samples. Similarly for CIFAR-10 dataset,

*CIFAR-10.* we random select initial training dataset with 5,000 samples, $\bar{\mathcal{D}}_{\text{pool}}$ with 5,000 samples, and validation dataset $\mathcal{D}_{\text{val}}$ with 5,000 samples.

## C.2    IMPLEMENTATION DETAILS ON THE EMPIRICAL EXAMPLE IN FIGURE 1

We show an empirical example in figure 1 to provide some intuition as to why BALANCE and Batch-BALANCE are effective in practice. We train a BNN with an imbalanced MNIST training subset that contains 28 images for each digit in [1-8] and 1 image for digits 0 and 9. The cross-entropy loss is reweighted to balance the training dataset during training. We obtain 200 posterior samples of BNN and use them to get the predictions on $\bar{\mathcal{D}}_{\text{pool}}$. We compute the Hamming distances for predictions all sample pairs and use these precomputed distances to plot the predictions with t-SNE (Van der Maaten & Hinton, 2008). The equivalence classes are approximated by farthest-first traversal algorithm (FFT) (Gonzalez, 1985). In figure 1, the equivalence classes are highly imbalanced. The ground truth $\bar{\mathcal{D}}_{\text{pool}}$ dataset labels represent the target hypotheses embedding. This figure highlights the scenario where the *equivalence class-based* methods, e.g. ECED and BALANCE are better than BALD.

## D    SUPPLEMENTAL EMPIRICAL RESULTS

In this section, we provide additional experimental details and supplemental results to demonstrate the competing algorithms.

### D.1    EFFECT OF DIFFERENT CHOICES OF HYPERPARAMETERS

We compare BALD and BALANCE with batch size $B = 1$ and different $K$'s on an imbalanced MNIST dataset which is created by removing a random portion of images for each class in the training dataset. figure 7 (a) shows that BALANCE performs the best with a large margin to the curve of BALD. Note that BALANCE with $K = 50$ is also better than BALD with $K = 100$.

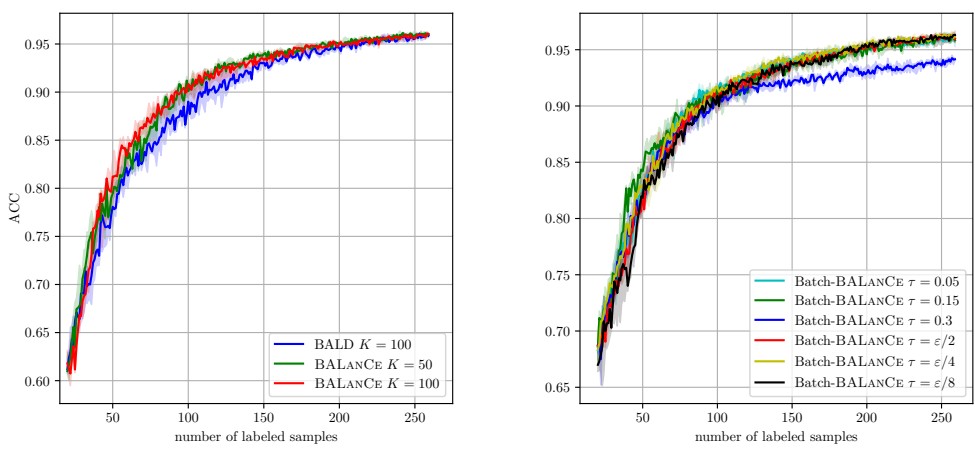

(a) ACC vs. # samples for different $K$'s.          (b) ACC vs. # samples for different $\tau$'s.

Figure 7: Learning curves of different $K$ and $\tau$ for BALANCE.

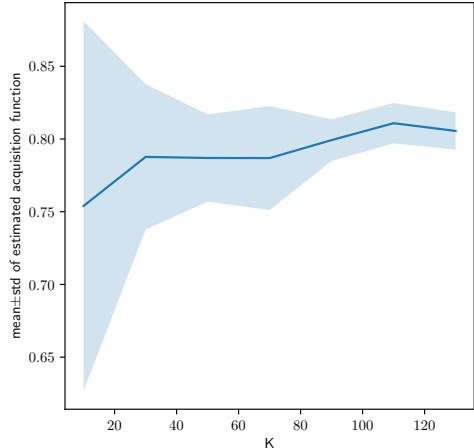

Figure 8: Estimated acquisition function values $\Delta_{\text{BALanCe}}$ of BALanCe vs. posterior sample number $K$

We also study the influence of $\tau$ for BALanCe on MNIST dataset. Denote the validation error rate of BNN model by $\varepsilon$. BALanCe with fixed $\tau = 0.05, 0.15, 0.3$ and annealing $\tau = \varepsilon/2, \varepsilon/4, \varepsilon/8$ are run on MNIST dataset and the learning curves are shown in figure 7 (b). The BALanCe is robust to $\tau$. However, when $\tau$ is set 0.3 and the test accuracy gets around 0.88, the accuracy improvement becomes slow. The reason for this slow improvement is that the threshold $\tau$ is too large and all the pairs of posterior samples are treated as in the same equivalence class and the acquisition functions for all the samples in the AL pool are zeros. In another word, the BALanCe degrades to random selection when $\tau$ is too large.

We further pick an data point from this imbalanced MNIST dataset and gradually increase the posterior sample number $K$ to estimate the acquisition function value $\Delta_{\text{BALanCe}}$ for this data point. For each posterior sample number $K$, we estimate the acquisition function $\Delta_{\text{BALanCe}}$ 10 times with 10 sets of posterior sample pairs. The mean and std for this $K$ are calculated and shown in figure 8.

### D.2    Experiments on other datasets

We compare different AL algorithms on tabular datasets including Human Activity Recognition Using Smartphones Data Set (Anguita et al., 2013) (HAR), Gas Sensor Array Drift (Vergara et al., 2012) (DRIFT), and Dry Bean Dataset (Koklu & Ozkan, 2020), as well as a more difficult dataset CINIC-10 (Darlow et al., 2018).

**HAR, DRIFT and Dry Bean Dataset**    We run 6 AL trials for each dataset and algorithm. In each iteration, the BNNs are trained with a learning rate of 0.01 and patience equal to 3 epochs. The BNNs all contain three-layer MLP with ReLU activation and dropout layers in between. The datasets are all split into starting training set, validation set, testing set, and AL pool. The AL pool is also used as $\mathcal{D}_{\text{pool}}$. The $\tau$ for Batch-BALanCe is set $\varepsilon/4$ in each AL loop. See table 2 for more experiment details of these 3 datasets.

| dataset | val set size | test set size | hidden unit # | sample # per epoch | K | B |
|---------|-------------|---------------|---------------|--------------------|----|----|
| HAR | 2K | 2,947 | (64,64) | 4,096 | 20 | 10 |
| DRIFT | 2K | 2K | (32,32) | 4,096 | 20 | 10 |
| Dry Bean | 2K | 2K | (8,8) | 8,192 | 20 | 10 |

Table 2: Experment details for HAR, DRIFT and Dry Bean Dataset

The learning curves of all 5 algorithms on these 3 tabular datasets are shown in figure 9. Batch-BALanCe outperforms all the other algorithms for these 3 datasets. For HAR dataset, both Batch-BALanCe and BatchBALD work better than random selection. In figure 9 (b) and (c), Mean STD, Variation Ratio and BatchBALD perform worse than random selection. We find similar effect for some other imbalanced datasets.

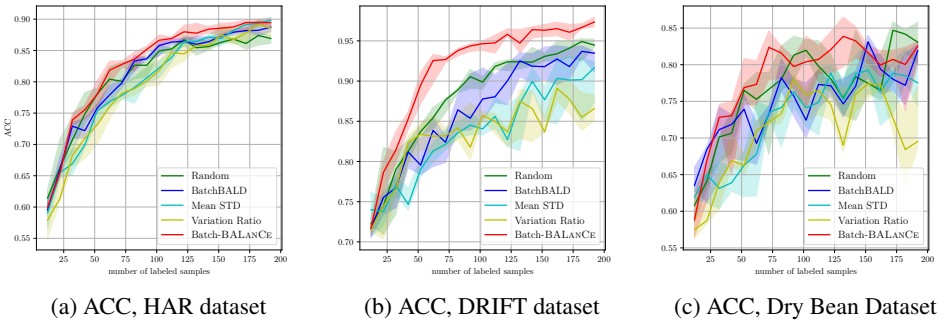

(a) ACC, HAR dataset  (b) ACC, DRIFT dataset  (c) ACC, Dry Bean Dataset

Figure 9: Experimental results on 3 tabular datasets. For all plots, the $y$-axis represents accuracy and $x$-axis represents the number of queried examples.

**CINIC-10** CINIC-10 is a large dataset with 270K images from two sources: CIFAR-10 (Krizhevsky et al., 2009) and ImageNet (Rasmus et al., 2015). The training set is split into an AL pool with 120K samples, 40K $\mathcal{D}_{\text{pool}}$ samples, 20K validation samples, and 200 starting training samples with 20 samples in each class. We use VGG-11 as the BNN. The number of sampled MC dropout pairs is 50 and the acquisition size is 10. We run 6 trials for this experiment. The learning curves of 5 algorithms are shown in figure 10. We can see from figure 10 that Batch-BALanCe performs better than all the other algorithms by a large margin in this setting.

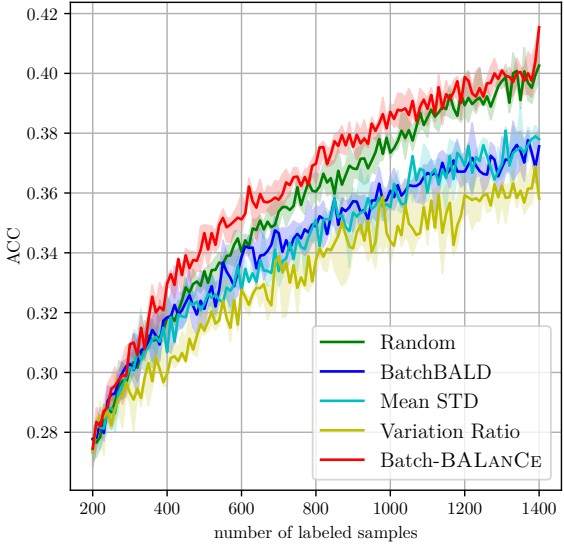

Figure 10: ACC vs. # samples on the CINIC-10 dataset.

**Repeated-MNIST with different amounts of repetitions** In order to show the effect of redundant data points on BathBALD and Batch-BALanCe, we ran experiments on Repeated-MNIST with

an increasing number of repetitions. The learning curves of accuracy for Repeated-MNIST with different repetition numbers can be seen in figure 11. A detailed model accuracy on the test dataset when the acquired training dataset size is 130 is shown in table 3. Even though Batch-BALANCE can improve data efficiency (Kirsch et al., 2019), there are still large gaps between the learning curves of Batch-BALD and Batch-BALANCE and the gaps become larger when the number of repetitions increases.

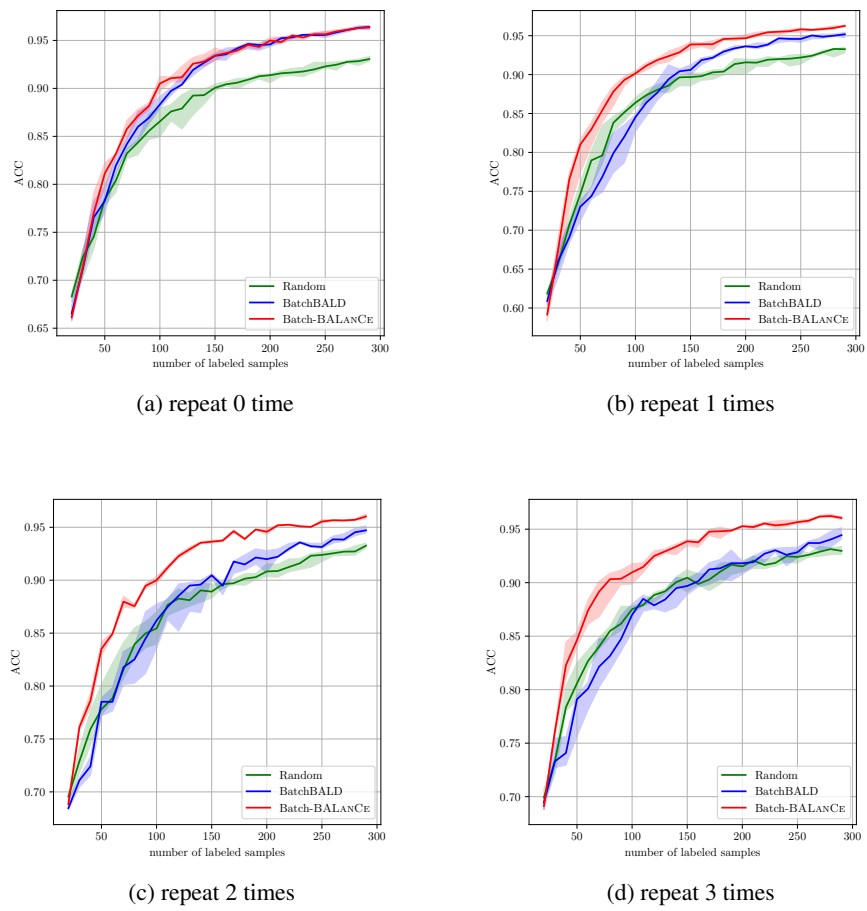

Figure 11: Performance of Random selection, BatchBALD, and Batch-BALANCE on Repeated-MNIST for an increasing number of repetitions. For all plots, the $y$-axis represents accuracy and $x$-axis represents the number of queried examples. We can see that BatchBALD also performs worse as the number of repetitions is increased. Batch-BALANCE outperforms BatchBALD with large margins and remains similar performance across different numbers of repetitions.

In order to compare our algorithms with other AL algorithms in this small batch size regime, we further run PowerBALANCE, PowerBALD, BADGE and CoreSet on the Repeated-MNIST with repeat number 3. As shown in figure 12, Batch-BALANCE achieves the best performance. Note that both PowerBALD and PowerBALANCE are efficient to select AL batch and show similar performance compared to BADGE algorithm.

**CIFAR-100** For CIFAR-100, we use 100 fine-grained labels. The dataset is split into initial training dataset with 5,000 samples, $\bar{\mathcal{D}}_{\text{pool}}$ with 5,000 samples, and validation dataset $\mathcal{D}_{\text{val}}$ with 5,000 samples. Experiment is conducted with batch size $B = 5,000$ and budget 25,000. The cSG-MCMC is used for BNN with epoch number 200, initial step size 0.5, and cycle number 4. We can see in figure 13 that both PowerBALANCE and Batch-BALANCE perform well in this dataset.

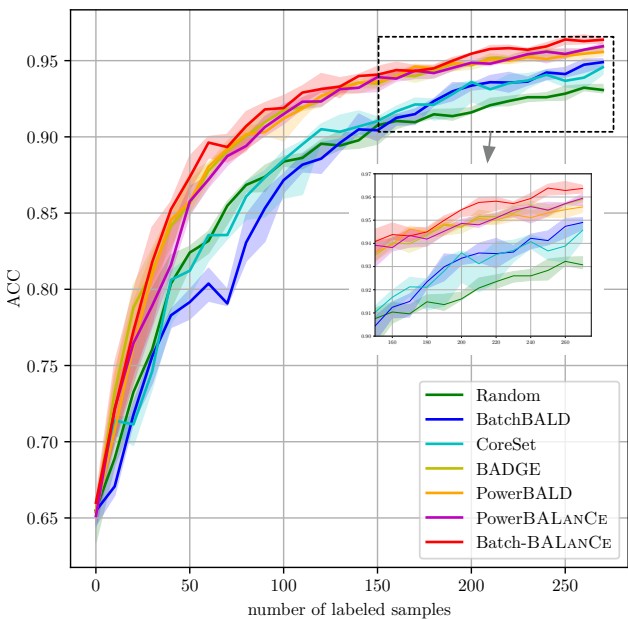

Figure 12: ACC vs. # samples on RepeatedMNIST dataset with repeat number 3.

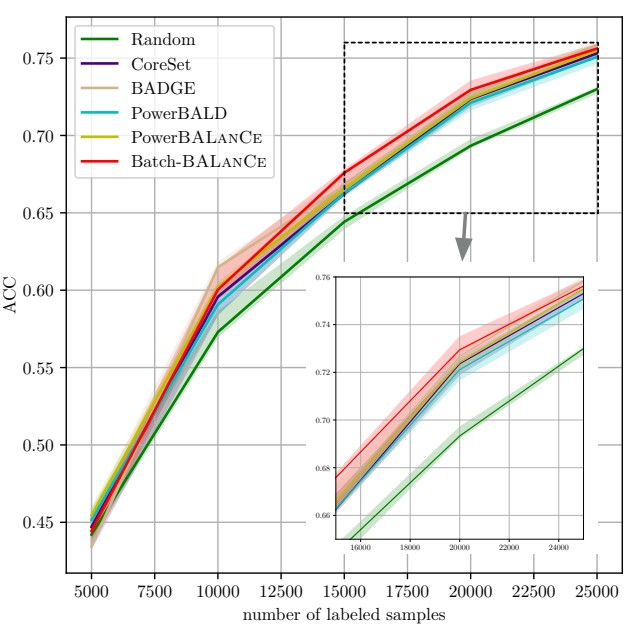

Figure 13: ACC vs. # samples, cSG-MCMC, CIFAR-100

### D.3 Additional evaluation metrics

Besides accuracy, we compared macro-average AUC, macro-average F1, and NLL for 5 different methods on EMNIST-Balanced and EMNIST-ByMerge datasets in figure 14. The acquisition size for all the AL algorithms is 5. Batch-BALanCe is annealed by setting $\tau = \varepsilon/4$. A macro-average AUC computes the AUC independently for each class and then takes the average. Both macro-average AUC and macro-average F1 take class imbalance into account. As shown in figure 14, Batch-BALanCe attains better data efficiency compared with baseline models on both balanced and imbalanced datasets.

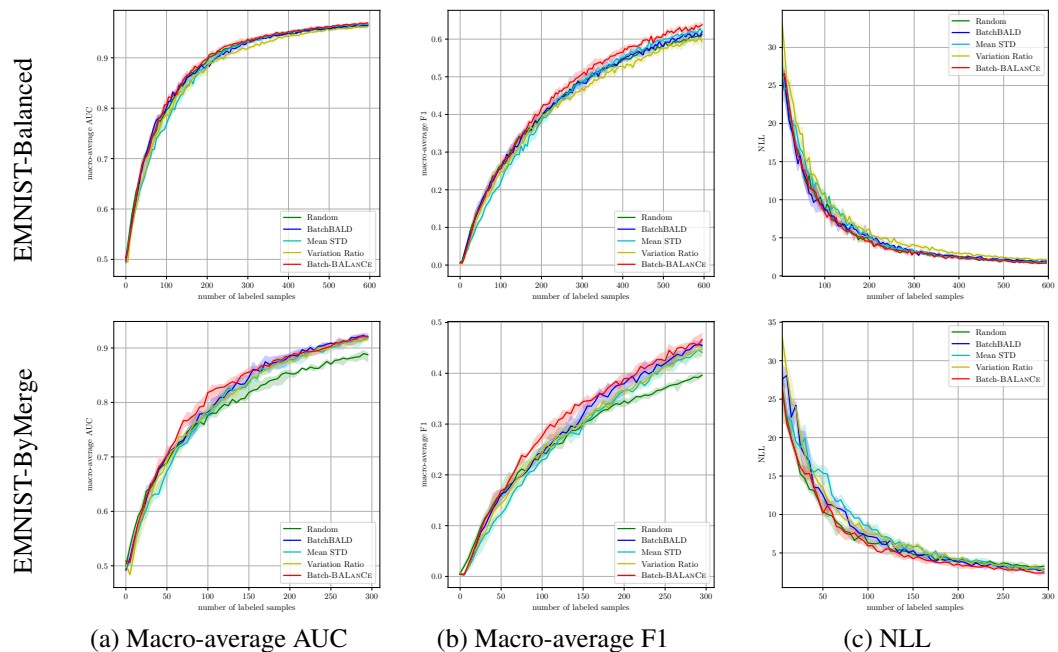

Figure 14: Compare different metrics for EMNIST-Balanced and EMNIST-Bymerge

We also evaluated the negative log-likelihood (NLL) for different AL algorithms. NLL is a popular metric for evaluating predictive uncertainty (Quinonero-Candela et al., 2005). As shown in figure 14, Batch-BALANCE maintains a better or comparable quality of predictive uncertainty over test data.

### D.4 BALANCE VIA EXPLICIT PARTITIONING OVER THE HYPOTHESIS POSTERIOR SAMPLES

Another way of estimating the acquisition function is to construct the equivalence classes explicitly first (e.g. by partitioning the hypothesis spaces into $k$ Voronoi cells via max-diameter clustering and calculate the weight discounts of edges that connect different equivalence classes. Intuitively, explicitly constructing equivalence classes may introduce unnecessary edges as two closeby hypotheses can be partitioned into different equivalence classes; therefore leading to an overestimate of the edge weight discounted. We call this algorithm BALANCE-Partition.

In order to compare with BALANCE and Batch-BALANCE, we sampled $K$ pairs of MC dropouts to estimate the acquisition function of BALANCE-Partition. All the representations of $2K$ MC dropouts on $\bar{\mathcal{D}}_{\text{pool}}$ are generated. We run FFT (Gonzalez, 1985) with Hamming distances and threshold $\tau$ on these representations to get approximated ECs. Each data point has at most $\tau$ Hamming distance to the corresponding cluster center. FFT is a 2-approx algorithm and the optimal solution with the same cluster number has cluster diameter $\geq \frac{\tau}{2}$. After equivalence classes are returned, BALANCE-Partition calculates the edges discounts of all edges that connect different equivalence classes and estimates the acquisition function values of each data sample in the AL pool.

Although a faster method that utilizes complete homogeneous symmetric polynomials (Javdani et al., 2014) can be implemented to estimate the acquisition function values for BALANCE-Partition, experiments in figure 15 show that BALANCE-Partition can not achieve better performance than BALANCE and increasing the MC dropout number does not improve performance significantly.

| Method | repeat 1 time | repeat 2 times | repeat 3 times | repeat 4 times |
|---|---|---|---|---|
| Random | $0.887 \pm 0.017$ | $0.883 \pm 0.012$ | $0.881 \pm 0.013$ | $0.895 \pm 0.009$ |
| BatchBALD | $0.917 \pm 0.005$ | $0.892 \pm 0.023$ | $0.883 \pm 0.025$ | $0.881 \pm 0.014$ |
| Batch-BALANCE | $0.926 \pm 0.008$ | $0.923 \pm 0.008$ | $0.929 \pm 0.004$ | $0.927 \pm 0.010$ |

Table 3: Mean±STD of test accuracies when acquired training set size is 130

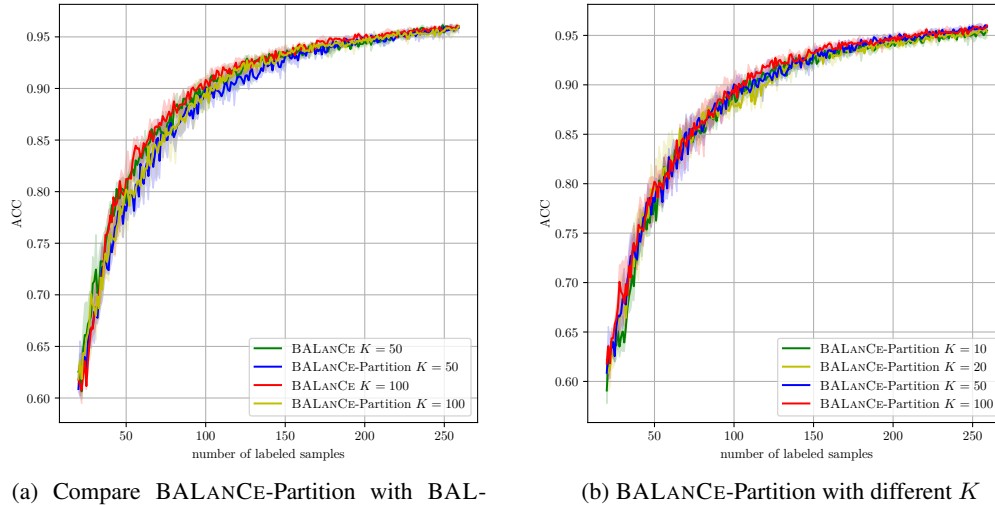

(a) Compare BALANCE-Partition with BALANCE

(b) BALANCE-Partition with different $K$

Figure 15: ACC vs. # samples for BALANCE-Partition and BALANCE.

### D.5 COEFFICIENT OF VARIATION

To gain more insight into why BALANCE and Batch-BALANCE work consistently better than BALD and BatchBALD, we further investigate the dispersion of the estimated acquisition function values for those methods. Since Batch-BALANCE and BatchBALD extend their fully sequential algorithms similarly in a greedy manner, we only compare the acquisition functions of BALANCE and BALD.

The coefficient of variation (CV) is chosen for the comparison of dispersion. It is defined as the ratio of the standard deviation to the mean. CV is a standardized measure of the dispersion of a probability distribution or frequency distribution. The value of CV is independent of the unit in which it is taken.

We conduct the experiment on the imbalanced MNIST dataset in the setting of appendix C.2. We estimate the acquisition function values of BALANCE and BALD 5 times with 5 sets of $K$ MC dropouts for each sample in the AL pool. Then, the CVs are calculated for these estimations. In figure 16, we show histograms of CVs for both methods. The estimated acquisition function values of BALANCE are less dispersed, which shows potential for better performance.

### D.6 PREDICTIVE VARIANCE

In order to directly compare the accuracy improvement of batches selected by different algorithms, instead of along the course of an AL trial, we conduct experiments with training sets of various sizes and compare the accuracy improvement of batches selected by AL algorithms with the same training set. The initial training set has 10 sampled randomly from Repeated-MNIST. In each step, we select 10 random samples and add them to training set. Hypotheses are drawn from BNN posterior given the current training set. We perform different AL algorithms and select batches with batch size 20. After each batch is added to training set, we can estimate the accuracy improvement of the batch. In each step, we perform each AL algorithm 20 times and estimate the mean and std of accuracy improvement. The mean and std of BNNs' accuracy are shown in figure 17. We can see in figure 17 that our algorithms consistently select batches that have high accuracy improvement and low variance.

### D.7 BATCH-BALANCE WITH MULTI-CHAIN CSG-MCMC

cSG-MCMC can be improved by sampling with multiple chains (Zhang et al., 2019). In order to evaluate different AL algorithms with this improved parallel cSG-MCMC method, we conduct

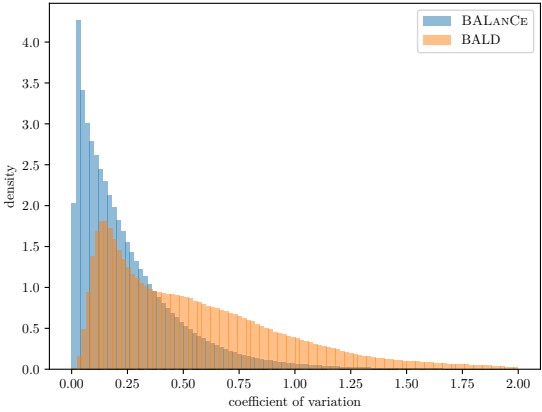

Figure 16: Histograms for coefficient of variation.

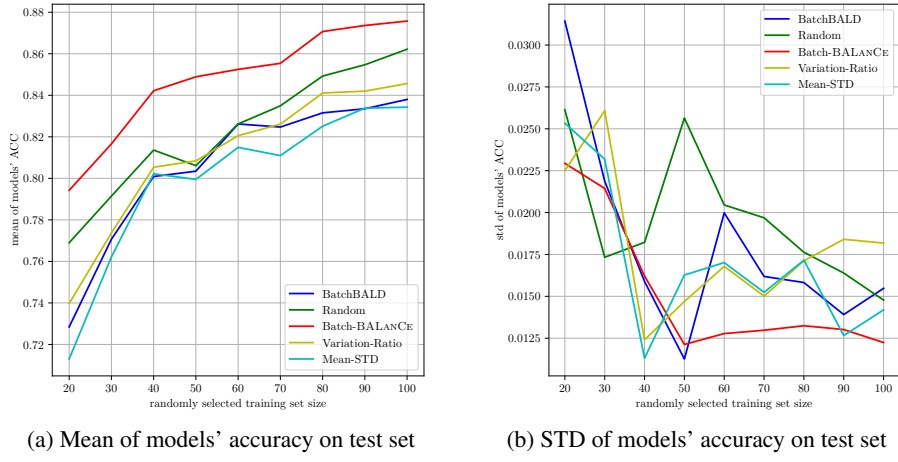

(a) Mean of models' accuracy on test set  (b) STD of models' accuracy on test set

Figure 17: We empirically show AL algorithms' predictive variance.

experiment on CIFAR-10 dataset with batch size $B = 5,000$. We sample posteriors with 3 chains. Each chain trains the model 200 epochs. The cycle number for each chain is 4 and 3 posterior samples are collected in each cycle. The result is shown in figure 18, Batch-BALANCE achieves better performance than BADGE.

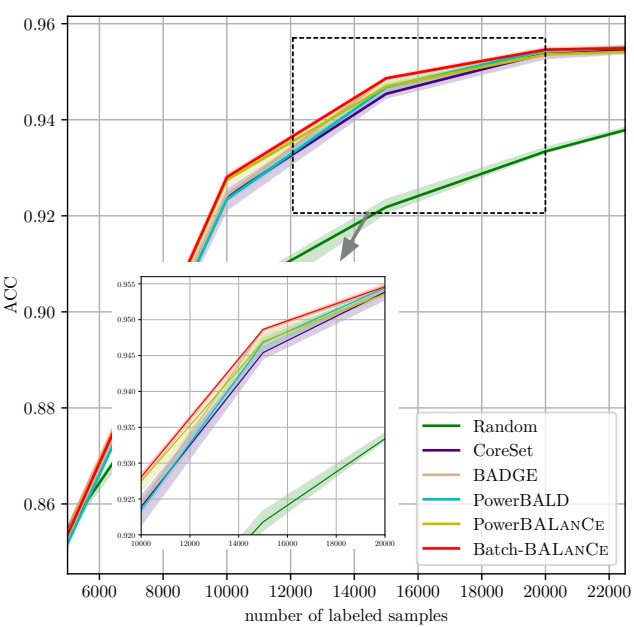

Figure 18: ACC vs. # samples, multi-chain cSG-MCMC, CIFAR-10

