# OpenReview forum: "Scalable Batch-Mode Deep Bayesian Active Learning via Equivalence Class Annealing"
_ICLR.cc/2023/Conference — ICLR 2023 poster_

### Official Review · Reviewer_CBRc · 2022-10-23

**Confidence:** 4
**Correctness:** 2
**Technical Novelty And Significance:** 2
**Empirical Novelty And Significance:** Not applicable
**Recommendation:** 8

**Clarity, Quality, Novelty And Reproducibility:**

I think the proposed AL method for BNN is novel and the motivation, intuition, as well as methodology, are clearly described in the paper. I have some concerns listed as follows:
The demonstration in figure 1 is not very clear to me. How the probability mass in figure b is computed given the EC is a set of N posterior distributions? How can we tell BALD samples are suboptimal from figure b?
Even though the proposed method does not require to construct ECs explicitly, I still think the number of ECs (i.e, k) is critical to the active sampling performance. The threshold \tau seems to reflect the number of k indirectly. Could the author further elaborate on the relationship between k and \tau and how should \tau be set intuitively?


**Strength And Weaknesses:**

Strength
The paper is well-written and well-motivated. I agree that the uncertainty estimation quality of deep neural networks, especially BNN poses a big challenge for modern active learning. I think the proposed method provides a new point of view for sampling based on the concept of equivalent classes. According to my understanding, the sampling strategy has the flexibility to put less focus on differentiating the hypothesis with little disagreement when the uncertainty estimation is not accurate in the early stage of active learning.
The proposed method is practical to real-world problems. The proposed active learning strategy avoids listing the full set of equivalence classes exhaustively by leveraging MC sampling.

Weakness
The paper lacks some theoretical discussion to draw the connection between the sampling strategy and learning performance. For example, what are the expected gain results from adding a new annotated sample selected by the proposed method? How fast would eq2 converge to eq1 as K increases?
The partition of the ECs is intuitive and lacks validation. Why is the Hamming distance used as a measurement and what are the alternative measures?


**Summary Of The Paper:**

This paper proposes a batch-mode active learning method for deep Bayesian Neural Networks. The sampling is based on the intuition of partitioning the equivalent classes optimally and puts less weight on the hypothesis that does not disagree with the true data distribution much.


**Summary Of The Review:**

The proposed method is novel and sound. But adding some theoretical underpinnings is prefered.

---

> ### Author Response · Authors · 2022-11-16
> **Responses to Reviewer CBRc (R4)**
>
> We thank the reviewer for the thoughtful comments! Below please find our detailed responses to the questions.
>
> ### 1) Theoretical justification and empirical validation of the proposed strategy
>
> ***Response.*** We agree with your interpretation of the intuition behind Batch-BALanCe! The theoretical analysis for deep (Bayesian) active learning is challenging in general; in this work, we can (theoretically) motivate the proposed algorithm from a **decision-theoretic standpoint** (as opposed to the uncertainty-sampling or diversity-sampling-based approaches):
>
> When one can *exactly infer the posterior* and/or *exhaustively enumerate the hypotheses*, the equivalence-class-based sampling/acquisition strategies (e.g., EC2, or ECED, as discussed in `section 2.2`) have been shown to have a **tight connection with the expected error rate** of the actively trained classifier. For instance, the sample complexity of the ECED strategy (see `appendix A.3`) is within a poly-log factor of the optimal algorithm that achieves some given target expected error rate. That is, the EC-based strategy is effective in improving the target performance. Our algorithm lifts those strict assumptions made for the canonical setting, and extends these insights to deep Bayesian AL.
>
> Given the above theoretical motivation, we further provide a stylized numerical example in `appendix A.1` to concretize this argument. Admittedly, as we mentioned in `section 6` (`Conclusion and Discussion`), a full-fledged analysis is out of the scope of this work. We instead resort to empirical validations of the proposed algorithm, in order to demonstrate that a novel adaptation of the EC-based approach works consistently well in the context of deep Bayesian AL. Concretely, in our experiments we show the average accuracy (among other metrics) of the model trained from examples from different methods. The expected gain of the proposed algorithm (i.e. the improvement in accuracy, or equivalently the reduction in error rate), as shown in `figure 3`, `4` in the main paper and `figure 6-9` and `11` in the appendix), robustly outperforms or matches the strongest baselines in consideration.
>
> ### 2) Convergence of $\Delta_{\text{BALanCe}}$ w.r.t. $K$ (Equation 2)
>
> ***Response.*** Thanks for this suggestion. In our original submission (`figure 6`, in `appendix D.1`), we have included empirical results on the robustness of our algorithm w.r.t. different K's. We now include an additional set of experiments in the revision to demonstrate the convergence of eq (2) as $K$ increases. These new results are provided in Appendix D.1 and Figure 8.
>
> ### 3) Motivation for Hamming distance
>
> ***Response.*** The choice of the distance measure has a decision-theoretic motivation. If the utility of choosing a hypothesis $h$, $\text{Utility}(h)$, is defined as the (validation) accuracy of $h$, then one can formulate the (active) learning problem as an *optimal value of information* problem:
>
> $$\min |\mathcal{D}_{\text{train}}|, \text{s.t.} \max _{\text{EC}} {\mathbb{E} _{h}[\text{Utility}(h) | \text{EC}, \mathcal{D}_\text{train}]} > Q$$
>
> where $\text{EC}$ denotes *equivalence class*, and $Q = 1 - \text{target error rate}$ is the target accuracy. In other words, we aim to identify the minimal training set, such that the expected accuracy of the resulting hypothesis (randomly sampled from the most likely equivalence class) is above the target accuracy.
>
> Therefore, **if the target performance metric is 0/1 loss**, then it is natural to define an equivalence class to be a collection of hypotheses that differ on at most $\tau$ fraction of labels, such that they have similar (validation) accuracy/error. We hence use the *Hamming distance* to construct ECs, which is a direct implication of the above formulation.
>
> Similarly, we can consider other distance functions if the goal is other than 0/1 loss (for example, one may consider a weighted distance measure if one gets penalized more on rare classes).
>
> ### 4) Clarification of the demonstration in figure 1
>
> ****Response.**** The probability of each EC is proportional to the number of posterior samples in that EC. The sampled posterior samples are heavily biased and cluttered towards suboptimal hypotheses (conceptually, this is similar to the numerical example provided in `figure 5` in the appendix).  In this case, the EC-based can be more effective.
>
> ### 5) The relationship between $k$ and $\tau$
> > Even though the proposed method ... how should $\tau$ be set intuitively?
>
> ****Response.**** Thanks for raising this important question. You are right that $\tau$ reflects the number of $k$: $\tau$ controls the largest diameter of the ECs; if we set $\tau$ smaller, the largest diameter becomes small and the number of ECs grows.
>
> We follow a standard hyperparameter tuning procedure and observe that $\tau\in [\text{validation error}/8,\text{validation error}/2]$ works generally well across all the datasets in our paper.

---

### Official Review · Reviewer_xKhG · 2022-10-27

**Confidence:** 3
**Correctness:** 3
**Technical Novelty And Significance:** 3
**Empirical Novelty And Significance:** 3
**Recommendation:** 8

**Clarity, Quality, Novelty And Reproducibility:**

Overall, the paper is clearly written making the proposed Bayesian AL method easy to follow and understand.
However, further improvement could be made as suggested in "Strength and Weakness".
The proposed method is relatively novel, rationally building on existing work on decision-theoretic AL schemes that adopt diversity sampling across equivalent classes and taking advantage of MC sampling schemes used for posterior estimation of BNNs.
The evaluation results clearly demonstrate the practical advantages and potential scalability of the proposed Batch-BALANCE algorithm.




**Strength And Weaknesses:**

OVERALL COMMENTS

Overall, the paper is well-written in a clear and logical manner.
The authors provide a brief yet informative review of the existing state-of-the-art relevant to the proposed batch AL algorithm, motivate and present their proposed approach, and assess the performance of the resulting AL algorithm Batch-BALANCE using several widely used benchmarks.
Evaluations based on different models and benchmarks with various batch sizes show that Batch-BALANCE generally outperforms other popular batch AL methods, including BatchBALD, BADGE, and PowerBALD.
Batch-BALANCE is shown to work well in both small and large batch regimes, providing effective and scalable means for Bayesian active learning.


DETAILED COMMENTS

1. The authors mention the downside of MIS (most informative selection) methods as "MIS tends to select data points that reveal the maximal information w.r.t. the sampled distribution, rather than guiding the active learner towards learning high accuracy models."
While this is a fair statement, the authors do not mention other AL schemes that focus on data acquisition that is expected to reduce model uncertainty that affects the model accuracy.
A well-known category is ELR (expected loss reduction), where an ELR-based Bayesian AL method tries to select unlabeled data points whose labeling is expected to reduce the loss most effectively.
It would be useful to briefly review representative and/or recent ELR-based Bayesian AL methods and discuss their relevance to the proposed algorithm, since at a high-level, they both take a "decision-theoretic" approach that focuses on the model accuracy.

2. Although the overall technical presentation in the paper is good, there are some minor suggestions for improving it further. For example:

- Please define ECED (page 4) before using the term, since it only appears in the appendix

- Please describe how the "Hamming distance" is computed between different hypotheses (or model parameters). Since Hamming distance is typically used to compare the difference between two symbol sequences by counting the number of locations with different symbols, it is not entirely clear how it is being used in this work for measuring the distance between two BNN parameters w and w'.

- While the authors present the definition of d_H in the appendix, it is still unclear how it applies to computing the distance between BNN parameters. Furthermore, it would be better if the definition of h_H is referred to in the main text.

- While the pseudocode of Algorithms 1-3 is helpful, it would make the pseudocode more readable for future readers if the code is (at least briefly) commented.

- Instead of calling I_Deltah_Balance(x,y) as a combinatorial information measure, it would be better to conceptually relate it to mutual information


3. The acquisition function of Batch-BALANCE is obtained through sampling using either MC dropout or  Stochastic gradient Markov Chain Monte Carlo (SG-MCMC).
However, there is currently no discussion on any pros/cons of the respective methods when used with Batch-BALANCE.





**Summary Of The Paper:**

This paper proposes a novel active learning (AL) algorithm for deep Bayesian neural networks (BNN).
Towards this goal, the paper proposes a novel acquisition function that is aimed at selecting data points that can improve the differentiation of models that belong to different equivalence classes.
By defining a decision-theoretic acquisition function that focuses on predictive accuracy and incorporating diversity sampling across equivalence classes, the paper shows that the resulting AL algorithm, referred to as Batch-BALANCE, is scalable and leads to AL performance improvement compared to existing alternatives.
.


**Summary Of The Review:**

This is a well-written paper that proposes a novel AL method that builds on a decision-theoretic principle and diversity sampling and aims to improve the performance and scalability of AL for deep BNN. The authors provide a brief review of relevant existing work to provide the right context for the proposed method, clearly motivating the work. Mostly, the technical details are elaborated clearly and the performance assessment results demonstrate the merits of the proposed Batch-BALANCE algorithm under a number of different AL scenarios.

. . .

The reviewer confirms that (s)he has reviewed the authors' rebuttal, which has been reflected in the above review comments and the overall evaluation scores

---

> ### Author Response · Authors · 2022-11-16
> **Responses to Reviewer xKhG (R3)**
>
> # Responses to Reviewer xKhG (R3)
>
> We thank the reviewer for the thoughtful comments! Below please find our detailed responses to the questions.
>
> ### 1) Review of ELR-based approaches
> > the authors do not mention other AL schemes that focus on data acquisition that is expected to reduce model uncertainty that affects the model accuracy. A well-known category is ELR (expected loss reduction), where an ELR-based Bayesian AL method tries to select unlabeled data points whose labeling is expected to reduce the loss most effectively. It would be useful to briefly review representative and/or recent ELR-based Bayesian AL methods and discuss their relevance to the proposed algorithm, since at a high-level, they both take a "decision-theoretic" approach that focuses on the model accuracy.
>
> ***Response.*** Thank you very much for the suggestion! We have added a brief discussion on ELR-based Bayesian AL in `section 5` (related work).
>
> ### 2) Other suggestions
>
> #### Define ECED before using the term.
> > Please define ECED (page 4) before using the term, since it only appears in the appendix
>
> ***Response.*** We have added a brief explanation of ECED to the main paper and further referred to `appendix A` for detailed explanation.
>
> #### Calculation of Hamming distance
> > Please describe how the "Hamming distance" is computed between different hypotheses (or model parameters). Since Hamming distance is typically used to compare the difference between two symbol sequences by counting the number of locations with different symbols, it is not entirely clear how it is being used in this work for measuring the distance between two BNN parameters $\omega$ and $\omega^{\prime}$. While the authors present the definition of $d_\mathrm{H}$ in the appendix, it is still unclear how it applies to computing the distance between BNN parameters. Furthermore, it would be better if the definition of $d_\mathrm{H}$ is referred to in the maintext.
>
> ***Response.*** Thanks for the suggestion! We have revised the paper accordingly. The distance $d_\mathrm{H}$ of between two hypotheses is Hamming distance between the their predictions on a randomly selected holdout dataset as explained in the 3rd paragraph of `section 1` (introduction).
>
> #### Comment of the pseudocodes
> > While the pseudocode of Algorithms 1-3 is helpful, it would make the pseudocode more readable for future readersif the code is (at least briefly) commented.
>
> ***Response.*** Comments about the pseudocodes are added in the revision.
>
> #### Relate $I_{\Delta_{BALanCe}}$ to mutual information
> > Instead of calling $I_{\Delta_{BALanCe}}(x,y)$ as a combinatorial information measure, it would be better to conceptually relate it to mutual information.
>
> Yes! $I_{\Delta_{BALanCe}}$ can indeed be interpreted as a similarity measure: as pointed out in `section 3.3`, our definition `eq (3)` was adapted from `Kothawade et al., 2021`, where the authors refer to $I_f(A,B)$ as submodular mutual information. Here we did not claim $\Delta_{BALanCe}$ to be a submodular function; nevertheless, we follow the similar insight to introduce $I_{\Delta_{BALanCe}}$ as a **goal-oriented similarity measure**.
>
> #### Pros/cons of MC dropout and cSG-MCMC
> > The acquisition function of Batch-BALANCE is obtained through sampling using either MC dropout or Stochastic gradient Markov Chain Monte Carlo (SG-MCMC). However, there is currently no discussion on any pros/cons of the respective methods when used with Batch-BALanCe.
>
> *****Response.***** Discussions of pros/cons of the MC dropout and cSG-MCMC are added to section 3.1 of the main paper. MC dropout is easy to implement and scales well to large models and datasets very efficiently. However, it is often poorly calibrated. cSG-MCMC is more practical and indeed has high fidelity to the true posterior. We want to point out that our findings are consistent across these two BNN sampling methods.

---

> > ### Comment · Reviewer_xKhG · 2022-11-17
> > **Thank you**
> >
> > Thank you for the detailed response.
> > It has addressed all concerns (mostly minor) regarding the original manuscript.

---

### Official Review · Reviewer_wGy4 · 2022-10-27

**Confidence:** 4
**Correctness:** 3
**Technical Novelty And Significance:** 3
**Empirical Novelty And Significance:** 3
**Recommendation:** 6

**Clarity, Quality, Novelty And Reproducibility:**

The article is written in a clear and consistent manner; the proposed method is an amalgamation of previously published works, but uses their results as an adaptation. Experimental study in different domains looks promising.

It is interesting to see how the proposed algorithms will work on CIFAR-100, which is considered a rather challenging dataset and requires large batches for active learning. It is also interesting to see how the proposed algorithms will behave when using deep ensembles to estimate uncertainty. While this is a more resource-intensive approach, it also yields more successful results compared to MC-dropout (Beluch et al. 2018).

Rather peculiar figure of the running time of the algorithms. We can see from Table 2 that PowerBALD shows top results with respect to alternatives with respect to computational complexity.  At the same time, in figure 2 this advantage is almost not visible due to the scale of the y-axis. I would suggest adding more points to this graph, or at least a zoom for the existing graph, similar to graph 4 (c). I think it is important to reflect this difference correctly, since at first glance PowerBALD on the graphs in most cases has comparable quality to the BALanCe algorithms, spending less computational resources.
Moreover, if on small datasets like Repeated-MNIST the difference in performance is noticeable, then on SVHN and CIFAR-10 it is practically not visible. It would also be interesting to see a comparison of BALanCE with PowerBALD and on small batches.
Also, it would be interesting to see a more extensive discussion of the results presented. Namely, why Power BALD has a comparable performance despite its disadvantages?
The same applies to the analysis of the complexity of the algorithms in Table 2 and the discussion on this.

**Strength And Weaknesses:**

Strengths:
- understandable and relevant motivation
- well and neatly written, consistent presentation of theory and results
- significant and detailed elaboration of related work, each used claim is supported
- the quality is presented in several metrics
- experiments are done in several domains

Weaknesses:
- Not clear why the particular criterion is used, i.e. the motivation behind the log-likelihood ration is not clear
- the usage of the particular distance (Hamming) is not explained - multiple distance can be considered
- importance sampling of configurations y_{1:b} was used and described in BatchBALD paper to calculate joint mutual information, so the novelty of adaptation of previous work on this step is questionable

**Summary Of The Paper:**

The authors presented a scalable batch Bayesian active learning algorithm and its different variations, which works equally well on large and small batches. The authors support their results with experiments on image datasets as well as with tabular data. By combining Hamming distance and combinatorial information measure, the algorithm is able to select samples for labelling based on both uncertainty and diversity, while spending a reasonable amount of computational time.

**Summary Of The Review:**

The paper is well written but the main technical part is too rushed - the design choices should be better explained. The results presented seem convincing. Generally, the paper in the current state is borderline, I current put it above the threshold, though very interested to see the authors response.

---

> ### Author Response · Authors · 2022-11-16
> **Responses to Reviewer wGy4 (R2)**
>
> # Responses to Reviewer wGy4 (R2)
> We thank the reviewer for the thoughtful comments! Below please find our detailed responses to the questions.
>
> ### 1) Motivation of likelihood ratio in equation (1)
> > Not clear why ... log-likelihood ratio is not clear
>
> ***Response.*** The likelihood ratio is used in Equation (1) (instead of the likelihood) so that the contribution of "non-informative examples" (e.g., when $\forall y_{1:b}^{\prime} \forall \omega, p(y_{1:b}^{\prime} | \omega, x_{1:b}) = \text{const}$ is zeroed out). We have revised the paper (`section 3.1`) to better motivate the acquisition function $\Delta_{\text{BALanCe}}(\cdot | \mathcal{D}_{\text{train}})$.
>
> ### 2) Motivation of Hamming distance
> > The usage of the particular distance is not explained - multiple distances can be considered
>
> ***Response.*** The Hamming distance between the two hypotheses reflects the **difference between prediction error**. The choice of the distance measure has a decision-theoretic motivation. If the utility of choosing a hypothesis $h$, $\text{Utility}(h)$, is defined as the (validation) accuracy of $h$, then one can formulate the (active) learning problem as an *optimal value of information* problem [`1`]:
>
> $$\min |\mathcal{D}_{\text{train}}|,  \text{s.t.} \max _{\text{EC}} {\mathbb{E} _{h}[\text{Utility}(h) | \text{EC}, \mathcal{D}_\text{train}]} > Q$$
>
> where $\text{EC}$ denotes *equivalence class*, and $Q = 1 - \text{target error rate}$ is the target accuracy. In other words, we aim to identify the minimal training set, such that the expected accuracy of the resulting hypothesis (randomly sampled from the most likely equivalence class) is above the target accuracy.
>
> Therefore, **if the target performance metric is 0/1 loss**, then it is natural to define an equivalence class to be a collection of hypotheses that differ on at most $\tau$ fraction of labels, such that they have similar (validation) accuracy/error. We hence use the *Hamming distance* to construct ECs, which is a direct implication of the above formulation.
> Indeed, different distances may be used, **if we aim at alternative performance measurements other than the prediction error (0/1 loss)**. For example, one may consider a weighted distance measure if one gets penalized more on rare classes.
>
> ### 3) On the novelty of the importance sampling
> > Importance sampling of configurations $y_{1:b}$ was ... previous work on this step is questionable
>
> ***Response.*** Thanks for this comment! For the small batch setting, the configuration sampling method is indeed inspired by `BatchBALD`. They also use importance sampling but the acquisition function to estimate is rather different, so we have to made adaptations accordingly, which led to rather different algorithmic and implementation details. We have added a footnote in the revision (`appendix B.2`) so that prior work is properly accredited.
>
> ### 4) Experiments on CIFAR-100
> > It is interesting to see how the proposed algorithms will work on `CIFAR-100`.
>
> ***Response.*** This is a great suggestion! We have added new experiments on `CIFAR-100` dataset, and observed similar behaviors of the competing algorithms as on the other datasets. Please refer to `appendix D.2` and `figure 13` for more details.
>
> ### 5) Experiments with deep ensembles
> > It is also interesting to see ... compared to MC-dropout.
>
> ***Response.*** Thanks for the suggestion! Deep ensemble method is an alternative to BNNs that is simple to implement, readily parallelizable, and requires very little hyperparameter tuning. Our paper focuses on Bayesian AL, and we agree that the ensemble method (sampling with multiple chains) could generate high-quality posteriors.
>
> We have evaluated the proposed methods with multi-chain cSG-MCMC that uses 3 chains (in parallel, can be seen as ensembles) to generate posterior samples. We have added these new experiments for multi-chain cSG-MCMC. Please refer to `appendix D.7` and `figure 18` for the results.
>
> ### 6) Computational analysis
> > Rather peculiar figure ...spending less computational resources.
>
> ***Response.*** We added `figure 6` to reflect the clear difference of the computational costs of AL algorithms. We want to point that our algorithm `PowerBALanCe` based on our new acquisition function and power sampling achieves better performance than `PowerBALD` (see `figure 4c`, `figure 13`) even though they share similar computation complexity (see `table 1`,  `figure 2` and `figure 6`). Detailed discussion of computational complexity is added to `appendix B.4`.
>
> ### 7) Comparison of BALanCe with PowerBALD and on small batches
> > Moreover, if on small datasets ... and the discussion on this.
>
> ***Response.*** Experiments on `Repeated-MNIST` with `PowerBALD`, `BADGE` and `CoreSet` are conducted and result is added to `figure 12` and `appendix D.2`. We can see from `figure 4c`, `13` that `PowerBALanCe` is actually better than `PowerBALD` and We can further improve it by `BALanCe-Clustering` algorithm.

---

> > ### Comment · Reviewer_wGy4 · 2022-12-13
> > **Thanks for the comments**
> >
> > Thanks for your comments and additional experiments. I still see the design choices a bit arbitrary. It might be interesting to do some additional study that validates them. I tend to keep my score but I increase my confidence in it.

---

> > > ### Author Response · Authors · 2022-12-13
> > > **Thanks for the further feedback**
> > >
> > > Thank you for your comments and for the discussion!
> > >
> > > We would like to briefly summarize our empirical study, in the hope that this could help clarify our design choice of the experiments.
> > >
> > > ---
> > >
> > > ### Ablation
> > >
> > > We have exhaustively evaluated how the choices of the hyperparameters affect the performance of our algorithm (i.e. parameter relevant to the construction of equivalence classes), measured by different metrics (e.g., downstream classification accuracy, or estimation accuracy in the proposed objective function, etc), as well as intermediate statistics useful for interpreting the final performance. We summarize the detailed ablation experiments as below.
> > >
> > > * In `Figures 7(a)`, we show *how choices of the posterior sample size* influence the performance. The posterior sample number is a common hyperparameter for AL algorithms in Bayesian settings. This figure shows how the approach would work on a large-scale real application with different hyperparameters.
> > > * In `Figures 7(b)`, we show *how choices of the EC threshold $\tau$ influence the performance*. Note that $\tau$ is the only hyperparameter specific to our algorithm. $\tau$ determines the size of ECs, so the plot shows how sensitive Batch-BALanCE is w.r.t. different ECs, and also justify that a proper choice of EC hyperparameter as elaborated in the paper is important to the AL performance.
> > > * In `Figures 1(a)` and `1(b)`, we show *what equivalence classes between hypotheses and the corresponding distribution are in a real experiment* (with formal definitions provided in `Section 2.2`)
> > > * In `Figure 8`, we empirically illustrate *how the estimated BALanCe acquisition function value (i.e., `eq (2)`) converges to the true BALanCe acquisition function value (i.e., `eq (1)`)*.
> > > * In `Table 1`, `Figure 2`, and `Figure 6`, we show the *computational complexity and computation time* for different AL algorithms w.r.t. batch size.
> > > * In `Figure 16`, we show that our algorithm based on ECs has *less dispersed estimations*.
> > > * In `Figure 15`, we show the influence of *different BALanCe acquisition function estimation strategies*, i.e., sampling hypothesis pairs and FFT (i.e, explicitly estimating ECs first and then calculating the expected weight discount of edges between ECs).
> > > * In `Figure 17`, we directly show the *predictive variance* of accuracy improvement of batches selected by different AL algorithms.
> > >
> > > ---
> > >
> > > ### Experimental setup
> > >
> > > In addition to the above study, we would like to briefly summarize the experiment settings our paper has explored to validate our design choice:
> > > * `dataset`: MNIST, RepeatedMNIST (with various repeat numbers), EMNIST (EMNIST-Balanced, EMNIST-ByMerge, EMNIST-ByClass), CINIC-10, CIFAR-10, CIFAR-100, SVHN, and UCI tabular datasets (HAR, DRIFT, Dry Bean Dataset).
> > > (*please note that our collection of datasets are a superset of those in many recent works, e.g., CoreSet, BADGE, and BatchBALD.*)
> > > * `metric`: ACC, Macro-average AUC, Macro-average F1, NLL
> > > * `batch size`: ranging from 1 to 5,000.
> > > * `BNN choices`: MC-dropouts, cSG-MCMC, and multi-chain cSG-MCMC.
> > > * `model architecture`: MLP, CNN with 3 blocks of [convolution, dropout, max-pooling, relu], VGG, and ResNet.
> > >
> > > ---
> > >
> > > We hope this could help clarify our design choices. If you have further questions, concerns, or suggestions, please do not hesitate to let us know. We are always happy to respond.

---

> ### Author Response · Authors · 2022-12-06
> **Follow-up Discussion**
>
> Dear Reviewer wGy4,
>
> We wanted to follow up to see if our responses adequately addressed the concerns you raised in your review of our paper. We would be very grateful if you could provide any additional feedback or comments you may have. We are happy to provide further clarifications or explanations if needed.
>
> Thank you very much for your time and consideration!
>
> Paper4976 Authors

---

### Official Review · Reviewer_RVjP · 2022-10-30

**Confidence:** 4
**Correctness:** 2
**Technical Novelty And Significance:** 3
**Empirical Novelty And Significance:** 3
**Recommendation:** 5

**Clarity, Quality, Novelty And Reproducibility:**

I think the paper is fairly clearly written and the approach is fairly novel. Yet the experiments are not very convincing.

**Strength And Weaknesses:**

Strengths: Overall I think the paper describes the problem of focus in a lot of detail in the introduction and clearly provides algorithms for each of the strategies used ie stochastic batch selection and greedy selection. The authors also compare the proposed approach to several active learning algorithms and variants of the balancing procedure in small and large batch settings.

Weaknesses: The overall aim of the paper is to use make active learning scalable by learning equivalence classes in settings where large batches of queries are common like many applications in reality. In the experiments, though the authors present several baselines on MNIST, EMNIST, CIFAR etc we never get a sense of how the approach would work on a large scale real application or what the challenges would be in a real setting. This is severe limitation. In fact,  the authors don't describe in any detail what the limitations of the proposed approach is.

Moreover, the focus is on learning and characterising equivalence classes with similar predictions but there is no analysis presented on what these equivalence classes between hypotheses are in the experimental section nor a sense of why these equivalence classes make sense. I would have liked to have seen some qualitative analysis of these.

A minor comment would also be to clearly label all axes. The authors state the axis labels in the caption but not consistently for all images which is confusing.


**Summary Of The Paper:**

Active learning  is widely used for selecting data efficiently for machine learning models. Performance of existing methods is largely dictated by the quality of uncertainty estimates of the model which may make it challenging to scale up to large batches. The authors propose a technique known as Batch Balance based on decision-theoretic active learning to characterise and distinguish between equivalence classes with similar predictions. They subsequently use a combinatorial information measure to define an acquisition procedure that enables the procedure to be scaled to large batches. Through several experiments the authors  try show their algorithm can effectively handle multi-class classification tasks, while having reasonable performance on both low and large batch settings.

**Summary Of The Review:**

Overall I recommend a weak reject of the paper mainly because there are many claims made in the motivation and introduction of the paper that are not demonstrated empirically.

---

> ### Author Response · Authors · 2022-11-16
> **Responses to Reviewer RVjP (R1)**
>
> # Responses to Reviewer RVjP (R1)
>
> We thank the reviewer for the thoughtful comments! Below please find our detailed responses to the questions.
>
> ### 1) Limitation of the proposed approach and challenges in realistic scenarios
> > In the experiments, though the authors present several baselines on MNIST, EMNIST, CIFAR, etc., we never get a sense of how the approach would work on a large-scale real application or what the challenges would be in a real setting. This is a severe limitation.
>
> ***Response.***  We would like to respectively argue that our experiments were grounded on a few recent works on **scalable active learning**, and that our proposed algorithm made strides towards more effective and efficient data acquisition strategies under realistic settings. More specifically, we conducted experiments on benchmarks datasets commonly used in the AL literature, including
> * a collection of UCI datasets (common for benchmarking AL on "small" statistical or probabilistic models),
> * MNIST, EMNIST, Fashion MNIST (common for benchmarking AL on deep models),
> * CIFAR, CINIC, and SVHN (common for benchmarking scalable AL algorithms, e.g., as studied in `Citovsky et al., 2021`).
>
> This collection of datasets extends beyond the datasets considered by the original works on CoreSet (`Sener & Savarese, 2017`), BADGE (`Ash et al., 2019`), BatchBALD (`Citovsky et al., 2021`).
>
> ---
>
> > In fact, the authors don't describe in any detail what the limitations of the proposed approach is.
>
> Thanks for raising this important question. As we lay out in the introduction, we aim to (1) mititgate the limitations of uncertainty-based deep AL heuristics due to **inaccurate uncertainty estimation**, and (2) enable **efficient computation of batches of queries** at scale. Although Batch-BALanCe demonstrates compelling empirical behaviors on both fronts, it is still a Bayesian algorithm and is limited by the quality of uncertainty estimates and the computational overhead of sampling, etc.
>
> In addition, as implied by our experimental results, there is a **trade-off between computational efficiency and model effectiveness**, even among variants of the proposed algorithms. We would like to point out that Batch-BALanCe shares similar computational complexity as CoreSet and BADGE when selecting large AL batches, i.e., they all have complexity quadratic w.r.t. $B$. While PowerBALD and PowerBALanCe (our AL algorithm based on BALanCe acquisition function and power sampling) are both efficient (as shown in `table 1`, `figure 2`, and the *newly added* `figure 6` in the revision), their performances are slightly worse for both small and large batch regime.
>
>
> ### 2) Qualitative analysis of Batch-BALanCe
> > Moreover, the focus is on learning and characterising equivalence classes with similar predictions but there is no analysis presented on what these equivalence classes between hypotheses are in the experimental section nor a sense of why these equivalence classes make sense. I would have liked to have seen some qualitative analysis of these.
>
> ***Response.*** In the 3rd paragraph of `section 1` and `appendix A.1`, we provide two concrete examples to illustrate why equivalence-class-based approaches can be better, both in the context of training deep neural network (`figure 1`, with experimental details provided in `appendix C.2`), and a stylized numerical example where the performance gap between equivalence-class-based approach and the uncertainty-sampling-based approach can be made large. Note that the results shown in `figure 1` is exactly the **intermediate qualitative results** (i.e. MC-dropout samples of NNs) underlying the performance plots on MNIST, as reported in `figure 3` in the `experiments` section.
>
> Furthermore, Batch-BALanCe **does not explicitly construct equivalence classes** to estimate the acquisition function. In `appendix D.4`, we explored approaches that estimate the acquisition function by *explicitly constructing* equivalence classes first with a farthest-first traversal procedure. The method, which is called `BALanCe-Partition`, does not have significant performance improvement and is more computationally expensive.
>
> We hope this clarifies the concern, and are happy to answer any follow-up questions!
>
>
> ### 3) Other comments
> > A minor comment would also be to clearly label all axes.
>
> ***Response.*** Thank you very much for the suggestion! We have labeled all axes in the revision.

---

> ### Author Response · Authors · 2022-12-06
> **Follow-up Discussion**
>
> Dear Reviewer RVjP,
>
> We wanted to follow up to see if our responses adequately addressed the concerns you raised in your review of our paper. We would be very grateful if you could provide any additional feedback or comments you may have. We are happy to provide further clarifications or explanations if needed.
>
> Thank you very much for your time and consideration!
>
> Paper4976 Authors

---

> ### Comment · Reviewer_RVjP · 2022-12-06
> **Follow-up Discussion**
>
> While I have read the author comments, i still dont feel all my questions have been adequately addressed particularly regarding my comment about equivalence classes. I understand the motivation for using equivalence classes and that the method doesnt explicitly construct them, but I still think its crucial to understand how the choice of equivalence classes influences performance. I am keeping my score as is.

---

> > ### Author Response · Authors · 2022-12-10
> > **Further clarification of this concern**
> >
> > Thank you for your feedback and for engaging in the discussion! Please see our clarification for the concern.
> >
> > We see two possible interpretations of the question
> > > how the choice of equivalence classes (ECs) influences performance?
> >
> > as elaborated below:
> >
> > ---
> >
> > If the question is on ``what's the effect of different hyperparameters of Batch-BALanCe (that lead to different ECs)'', we would like to point out a few existing results reported in the paper that directly answer this question:
> > * In `Figures 7(a)`, we show *how choices of the posterior sample size* influence the performance. The posterior sample number is a common hyperparameter for AL algorithms in Bayesian settings. This figure shows how the approach would work on a large-scale real application with different hyperparameters.
> > * In `Figures 7(b)`, we show *how choices of the EC threshold $\tau$ influence the performance*. Note that $\tau$ is the only hyperparameter specific to our algorithm. $\tau$ determines the size of ECs, so the plot shows how sensitive Batch-BALanCE is w.r.t. different ECs, and also justify that a proper choice of EC hyperparameter as elaborated in the paper is important to the AL performance.
> >
> > In addition to the above sensitivity analysis, we have performed both *qualitative* and *quantitative* studies on justifying ECs are a *useful* concept for Bayesian deep active learning. To highlight a few that are relevant:
> >
> > * In `Figures 1(a)` and `1(b)`, we show what equivalence classes between hypotheses and the corresponding distribution are in a real experiment (with formal definitions provided in `Section 2.2`)
> > * In `Figure 8`, we empirically illustrate how the estimated BALanCe acquisition function value (i.e., `eq (2)`) converges to the true BALanCe acquisition function value (i.e., `eq (1)`).
> > * In `Table 1`, `Figure 2`, and `Figure 6`, we show the computational complexity and computation time for different AL algorithms w.r.t. batch size.
> > * In `Figure 16`, we show that our algorithm based on ECs has less dispersed estimations.
> > * In `Figure 15`, we show the influence of different BALanCe acquisition function estimation strategies, i.e., sampling hypothesis pairs and FFT (i.e, explicitly estimating ECs first and then calculating the expected weight discount of edges between ECs).
> > * In `Figure 17`, we directly show the predictive variance of accuracy improvement of batches selected by different AL algorithms.
> >
> > ---
> >
> > If by "*how the choice of equivalence classes influences performance*", the reviewer is referring to whether or not including ECs is a critical factor for the algorithm's performance, our answer is yes. We demonstrate the compelling and consistent performance of Batch-BALanCe compared to non-EC-based approaches through extensive experiments:
> >
> > * dataset: MNIST, RepeatedMNIST (with various repeat numbers), EMNIST (EMNIST-Balanced, EMNIST-ByMerge, EMNIST-ByClass), CINIC-10, CIFAR-10, CIFAR-100, SVHN, and UCI tabular datasets (HAR, DRIFT, Dry Bean Dataset).
> > * metric: ACC, Macro-average AUC, Macro-average F1, NLL
> > * AL batch size: ranging from 1 to 5,000.
> > * BNN choices: MC-dropouts, cSG-MCMC, and multi-chain cSG-MCMC.
> > * model architecture: MLP, CNN with 3 blocks of [convolution, dropout, max-pooling, relu], VGG, and ResNet.
> >
> > Our EC-based approach does show a significant advantage over non-EC-based approaches and EC indeed is a critical factor for our EC-based algorithm.
> >
> > To further concretely address the reviewer's concern about the choice of ECs, we summarize an additional collection of statistics from the experiment of figure 3(a) where we show the `estimated BALanCe acquisition function value` of batches selected by various AL algorithms.
> >
> >
> > |labeled samples number|20  |30  |40  |50  |60  |70  |80  |90  |
> > |----------------------|-----|-----|-----|-----|-----|-----|-----|-----|
> > |BALD; BALanCe acquisition|0.928$\pm$ 0.018|0.929$\pm$ 0.019|0.890$\pm$ 0.044|0.899$\pm$ 0.029|0.923$\pm$ 0.016|0.905$\pm$ 0.032|0.905$\pm$ 0.017|0.904$\pm$ 0.024|
> > |BALD; ACC|0.627$\pm$ 0.019|0.694$\pm$ 0.021|0.737$\pm$ 0.021|0.780$\pm$ 0.027|0.813$\pm$ 0.031|0.843$\pm$ 0.021|0.859$\pm$ 0.008|0.878$\pm$ 0.016|
> > |Random; BALanCe acquisition|0.355$\pm$ 0.302|0.291$\pm$ 0.265|0.447$\pm$ 0.285|0.345$\pm$ 0.349|0.070$\pm$ 0.105|0.207$\pm$ 0.199|0.347$\pm$ 0.321|0.203$\pm$ 0.254|
> > |Random; ACC|0.618$\pm$ 0.020|0.680$\pm$ 0.031|0.716$\pm$ 0.027|0.754$\pm$ 0.030|0.800$\pm$ 0.040|0.812$\pm$ 0.025|0.831$\pm$ 0.026|0.845$\pm$ 0.018|
> > |Batch-BALanCe; BALanCe acquisition|**0.948$\pm$ 0.006**|**0.942$\pm$ 0.008**|**0.942$\pm$ 0.012**|**0.938$\pm$ 0.007**|**0.942$\pm$ 0.009**|**0.930$\pm$ 0.008**|**0.938$\pm$ 0.009**|**0.935$\pm$ 0.013**|
> > |Batch-BALanCe; ACC|**0.634$\pm$ 0.017**|**0.697$\pm$ 0.013**|**0.748$\pm$ 0.021**|**0.793$\pm$ 0.016**|**0.827$\pm$ 0.010**|**0.844$\pm$ 0.016**|**0.864$\pm$ 0.022**|**0.882$\pm$ 0.013**|

---

> > ### Author Response · Authors · 2022-12-10
> > **Further clarification of this concern (second part)**
> >
> > For convenience, we highlight the best-performing scores in **bold**. We can see that Batch-BALanCe, which is designed to optimize the BALanCe acquisition function, not only achieves high *BALanCe acquisition function values* but also get high *model accuracies*. More importantly, the BALanCe acquisition function values of batches selected by different AL algorithms and the corresponding model target accuracies highly align with each other. Therefore, we believe the BALanCe acquisition function based on ECs is a strong indicator of the model's accuracy improvement.
> >
> > ---
> >
> > If you have any further questions, concerns, or suggestions, please do not hesitate to let us know. We are always open to feedback and discussion.
> >
> > Best,\
> > The authors of Paper4976

---

### Author Response · Authors · 2022-11-16
**Response to all reviewers**

We thank all the reviewers for their detailed reviews and constructive comments! Following these suggestions, we have further revised the paper which we hope to address most of the existing concerns. In particular, we have added references of ELR-based approaches, and included new experiment results with additional algorithmic variants, i.e., with parallel sampling method (`ensemble method`) and larger datasets (i.e. `CIFAR-100` dataset). We also added detailed discussions of computational complexity in the appendix. Revised texts are now highlighted in pink.

We provide more detailed responses to each reviewer following each official review, and are happy to answer any follow-up questions!

---

### Decision · Program_Chairs · 2023-01-20

**Decision:**

Accept: poster

**Justification For Why Not Higher Score:**

I find that missing comparison to margin sampling, which is often competitive in many AL scenarios, a weakness.  Apart from intuition, there isn't much in terms of theoretical justification for the suggested appraoch.

**Justification For Why Not Lower Score:**

After discussion and revision by the authors, reviewers tend to agree that there is enough empirical evidence to accept the paper.

**Metareview: Summary, Strengths And Weaknesses:**

This work presents a novel batch active learning algorithm set in a Bayesian framework, focused on training Bayesian Neural Networks (BNN).  The algorithm is motivated by labeling examples that are most useful in differentiating hypotheses from different equivalence classes, while reducing labels wasted on differentiating similar hypotheses. Furthermore, the authors provide a practical algorithm that scales efficiently with batch size by first clustering the sampling pool and executing independent sampling within each cluster. The empirical evaluation considers small (e.g. MNIST) and larger (e.g. SVHN) with small (e.g. 10) and larger (e.g. 1000) batch sizes, demonstrating improvements or comparable performance with a couple Baysian and non-Baysian active learning baselines in some cases with significantly less computation.

The majority of reviewers agree the paper provides a novel approach presented clearly with convincing enough empirical backing.  In the rebuttal the authors answer or clarify several concerns that were raised, including providing an additional more challenging empirical benchmark (CIFAR-100).  The strongest remaining reviewer comment is regarding the lack of a stronger theoretical motivation for the (implicit) equivalence hypothesis classes used to motivate the sampling approach.  However, given the contributions and appreciating the use of equivalence classes as a heuristic, I will recommend a weak accept.

I strongly encourage authors to incorporate all clarification into the final work.  Also, although not raised by reviewers, the AC strongly recommends including the simple but effective uncertainty (e.g. margin) sampling baseline to be included in the paper. Although simple and not tailored to the Bayesian setting necessarily, uncertainty sampling often provides strong performance.


**Note From Pc:**

if the above contains the word "oral" or "spotlight" please see: "oral" presentation means -> notable-top-5% and "spotlight" means -> notable-top-25%. As stated in our emails, we are disassociating presentation type from AC recommendations